# Benchmarking and Enhancing VLM for Compressed Image Understanding

**Zifu Zhang** [1 2]  **Tongda Xu** [1]  **Siqi Li** [1 3]  **Shengxi Li** [2]  **Yue Zhang** [2]  **Mai Xu** [2]  **Yan Wang** [1]

## Abstract

With the rapid development of Vision-Language Models (VLMs) and the growing demand for their applications, efficient compression of the image inputs has become increasingly important. Existing VLMs predominantly digest and understand high-bitrate compressed images, while their ability to interpret low-bitrate compressed images has yet to be explored by far. In this paper, we introduce the first comprehensive benchmark to evaluate the ability of VLM against compressed images, varying existing widely used image codecs and diverse set of tasks, encompassing over one million compressed images in our benchmark. Next, we analyse the source of performance gap, by categorising the gap from a) the information loss during compression and b) generalisation failure of VLM. We visualize these gaps with concrete examples and identify that for compressed images, only the generalization gap can be mitigated. Finally, we propose a universal VLM adaptor to enhance model performance on images compressed by existing codecs. Consequently, we demonstrate that a single adaptor can improve VLM performance across images with varying codecs and bitrates by 10%-30%. We believe that our benchmark and enhancement method provide valuable insights and contribute toward bridging the gap between VLMs and compressed images. The source code is available at https://github.com/bblgbr/CompressVLMBench.

## 1. Introduction

The boom of multimedia services and applications has resulted in dramatic increase in image data, creating significant challenges in terms of transmission bandwidth and storage capacity. To address this, efficient image compression methods are essential for reducing data volume while maintaining or enhancing the subjective visual quality for human perception. Over the past few decades, numerous advanced compression standards have been introduced, such as JPEG (Wallace, 1991), and VTM (Bross et al., 2021), alongside recent end-to-end learned compression approaches (Ballé et al., 2018; Lu et al., 2019; Cheng et al., 2020; Zhang et al., 2025b) and generative compression methods (Mentzer et al., 2020; Li et al., 2024c; Zhang et al., 2025a).

In parallel, the advent of Big Data has transformed the way intelligent machines interact with the world, leading to extensive research into compression techniques for machine vision tasks, as opposed to human vision. Notable examples include image coding for machines (ICM) (Kang et al., 2022) and feature coding for machines (FCM) (Rosewarne, 2023), emerging standards introduced by the MPEG. However, previous works (Kim et al., 2023; Zhang et al., 2024b; Chen et al., 2023a; Liu et al., 2026b) have largely focused on specific computer vision tasks, such as object detection and instance segmentation, using fixed backbone networks, which limits their generalization capabilities.

With the continuous advancements in the multimodal field, VLMs have developed rapidly (Chen et al., 2025; 2024). Current VLMs are not only capable of understanding complex images (Yao et al., 2024) but also performing tasks such as detection and segmentation (Feng et al., 2025), further generalizing and unifying visual tasks. This makes VLMs a promising and important direction for future development. To enhance the ability of VLMs to process compressed images, one approach is to optimize existing codecs to minimize the bitrate without compromising VLM performance. Research based on VCM and FCM has demonstrated good results when transferring tasks to VLM vision (Kao et al., 2024; Li et al., 2024a). However, these methods are specific to a particular codec or VLM, limiting the generalization capabilities. On the other hand, another approach is to improve the VLM's capacity to understand compressed images from various codecs, independent of specific compression distortions. This approach could enhance generalization, but to date, no research has investigated this area. Furthermore, although several benchmarks exist to evaluate VLMs' performance in tasks such as VQA (Hudson & Manning,

[1]Institute for AI Industry Research, Tsinghua University, Beijing, China [2]Beihang University, Beijing, China [3]Beijing University of Technology, Beijing, China. Correspondence to: Tongda Xu <x.tongda@nyu.edu>, Shengxi Li <LiShengxi@buaa.edu.cn>, Yan Wang <wangyan@air.tsinghua.edu.cn>.

*Proceedings of the 43rd International Conference on Machine Learning*, Seoul, South Korea. PMLR 306, 2026. Copyright 2026 by the author(s).

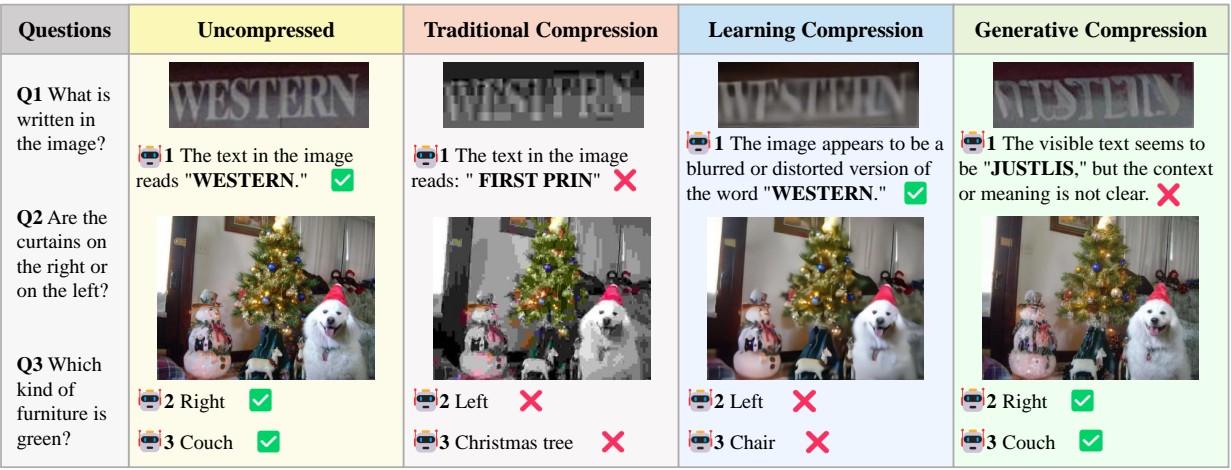

*Figure 1.* Comparative visualization of four image compression technique: uncompressed, traditional codec (JPEG), learning-based codec (ELIC), and generative codec (StableCodec), highlighting their impact on visual clarity and semantic preservation through targeted question-answering. All forms of compression-induced distortion affect the ability of VLMs to understand images.

2019), spatial relations (Li et al., 2023a), text understanding (Liu et al., 2024b), and knowledge answering (Yue et al., 2024), no benchmark has been developed specifically for assessing VLM performance on compressed images.

In this paper, we present a comprehensive benchmark designed to evaluate the ability of VLMs to understand and process compressed images. Our benchmark includes 11 widely-used codecs and 3 series of VLMs, ranging from 1 to 32 billion parameters, and assesses both coarse-grained and fine-grained metrics across more than 1 million compressed images. This large-scale analysis allows us to quantify the performance degradation of VLMs due to image compression, revealing that compression can significantly impair the model's ability to interpret visual content, as reflected in the subjective examples in Figure 1. Based on this, we further break down the observed performance gap into two distinct components: the information gap during compression, which directly impacts the fidelity of image features, and the generalization gap of VLMs, which limits their ability to adapt to compressed images. Through visualizations and examples, we show that while the loss of information during compression is inherent and cannot be easily mitigated, the generalization gap represents a gap that can be addressed by improving the model's ability to handle compressed inputs. To close this generalization gap, we propose a lightweight VLM adaptor that enhances VLM performance on compressed images across diverse codecs and bitrate levels. Empirical results demonstrate that the proposed adapter consistently improves compressed-image understanding by 10%–30% under different compression settings, indicating its strong generalization capability across codec types and compression levels. These results suggest that our method provides a practical and scalable solution for real-world VLM applications involving compressed visual inputs.

## 2. Related Works

**Image Coding for Humans.** In recent years, deep learning has significantly advanced image compression, surpassing traditional hand-crafted codecs. Existing methods can be grouped into three categories: traditional, learning-based, and generative compression. Traditional codecs, from JPEG (Wallace, 1991) to HEVC (Sullivan et al., 2012) and VVC (Bross et al., 2021), have steadily improved but at the cost of increasing computational complexity, eventually reaching a plateau. Learning-based compression methods began to emerge and have progressively incorporated state-of-the-art network architectures, such as ResNet (Cheng et al., 2020), Transformers (Liu et al., 2023a) and Mamba (Zeng et al., 2025), as well as entropy modeling along both channel and spatial dimensions (Minnen et al., 2018; He et al., 2022), achieving superior compression performance compared to traditional codecs. Nevertheless, due to their pixel-level optimization, learning-based methods often produce poor visual quality at extremely low bitrates, which is undesirable for human perception. To address this, generative model-based approaches, including those leveraging GANs (Mentzer et al., 2020; Muckley et al., 2023) and Diffusion models (Li et al., 2024c; 2025), have been developed to optimize compression with respect to perceptual quality. However, they remain computationally intensive and large in size, limiting practical deployment and leaving considerable room for improvement.

**Image Coding for Machines.** With the rapid development of computer vision and the widespread adoption of intelligent tasks, compression techniques tailored for machine vision have been extensively investigated. In response, the Moving Picture Experts Group (MPEG) established Video Coding for Machines (VCM) (Kang et al., 2022) and Feature

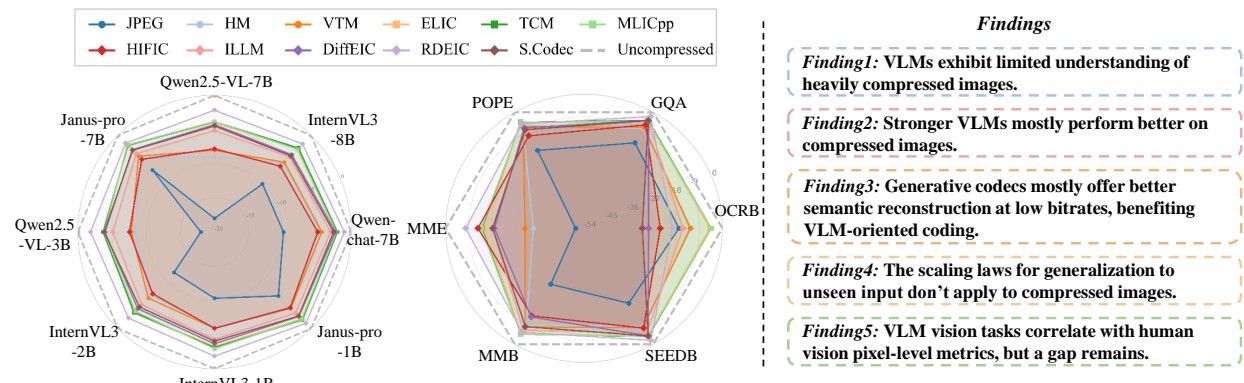

*Figure 2.* (a) BD-Metric values of different VLMs across different codecs for the same tasks (SEEDBench). (b) BD-Metric values for different tasks under various compression methods based on the same VLM (Qwen-VL2.5-3B). (c) Summary of our main findings.

Coding for Machines (FCM) (Rosewarne, 2023). Existing methods (Zhang et al., 2024b; Liu et al., 2023b; Zhang et al., 2024a; Li et al., 2024b) mainly focus on specific architectures, such as Fast-RCNN (Ren et al., 2015) and Mask-RCNN (He et al., 2017), and target particular tasks like object detection, instance segmentation, and object tracking, with limited generalization capabilities. Furthermore, driven by the recent success of VLMs and their increasingly broad applications, some studies (Kao et al., 2024; Li et al., 2024a) have begun exploring image compression for VLMs. However, these works typically focus on a single codec and lack a comprehensive benchmark like image compression for human vision (Hu et al., 2021; Liu et al., 2026a).

**Vision-Language Models.** VLMs are large-scale models that integrate visual modalities with language understanding (Zhan et al., 2024; Chen et al., 2025). In recent years, there has been a surge in research utilizing VLMs (Wang et al., 2024; Team et al., 2025) for tasks such as image understanding, image recognition, instance segmentation, and object detection, significantly enhancing the model's generalization capabilities. To evaluate the comprehensive performance of VLMs, several benchmark studies have been proposed, including SEEDBench (Li et al., 2023a), MM-Bench (Liu et al., 2024a), MME (Chaoyou et al., 2023), OCRBench (Liu et al., 2024b) and et al. However, the existing evaluation datasets consist of high-bitrate, clear images, and there has been little exploration of methods for evaluating VLM performance on compressed images, which are of great importance as they can significantly save bandwidth.

## 3. Benchmark Design

### 3.1. Overview

**Image Codecs.** We selected 11 state-of-the-art codecs with three representative categories, including traditional codecs, learning-based codecs, and generative codecs, as shown in Table 1. Specifically, for traditional codecs, we

chose the widely used JPEG (Wallace, 1991), HM (Sullivan et al., 2012) and VTM (Bross et al., 2021). For learning-based codecs, we selected commonly used ELIC (He et al., 2022), TCM (Liu et al., 2023a), and MLICpp (Jiang et al., 2023). For generative codecs, we selected the GAN-based HiFiC (Mentzer et al., 2020) and MS-ILLM (ILLM) (Muckley et al., 2023), as well as the diffusion-based DiffEIC (D.EIC)(Li et al., 2024c), RDEIC (Li et al., 2025), and StableCodec (S.Codec) (Zhang et al., 2025a). All compression methods were applied to the original dataset with four different bitrate levels to cover a wide range of bitrates. Detailed parameter settings are provided in the Appendix A.

*Table 1.* Assessing 9 VLMs of varying scales and 11 representative image codecs with 7 metrics.

| Codecs (11) | | Tasks (7) | | VLMs (9) | |
|---|---|---|---|---|---|
| Traditional | JPEG, HM, VTM | Coarse-grained | POPE, GQA, COCO-Caption | 1-3B | InternVL3, Janus-Pro, Qwen2.5-VL |
| Learning-based | ELIC, TCM, MLICpp | Fine-grained | OCRBench | 7-8B | Qwen-Chat, Qwen2.5-VL, Janus-Pro, InternVL3 |
| Generative | HiFiC, MS-ILLM, DiffEIC, RDEIC, StableCodec | Compre-hensive | MMB, MME, SEEDBench | 32B | Qwen2.5-VL |

**Datasets and Tasks.** To assess the impact of image compression on VLMs, we curate seven tasks covering coarse-grained and fine-grained evaluation. Coarse-grained tasks focus on semantic understanding, evaluated using the POPE (Li et al., 2023b), GQA (Hudson & Manning, 2019) and COCO-Caption (COCOC) (Chen et al., 2015) benchmarks, which encompass common vision-language tasks such as visual question answering, spatial reasoning, and image captioning. For fine-grained tasks, we use OCRBench (OCRB) (Liu et al., 2024b), comprising 1000 images across various resolutions, allowing us to test codec adaptability to different image sizes. Additionally, we include three comprehensive benchmarks: MMBench (MMB) (Liu et al.,

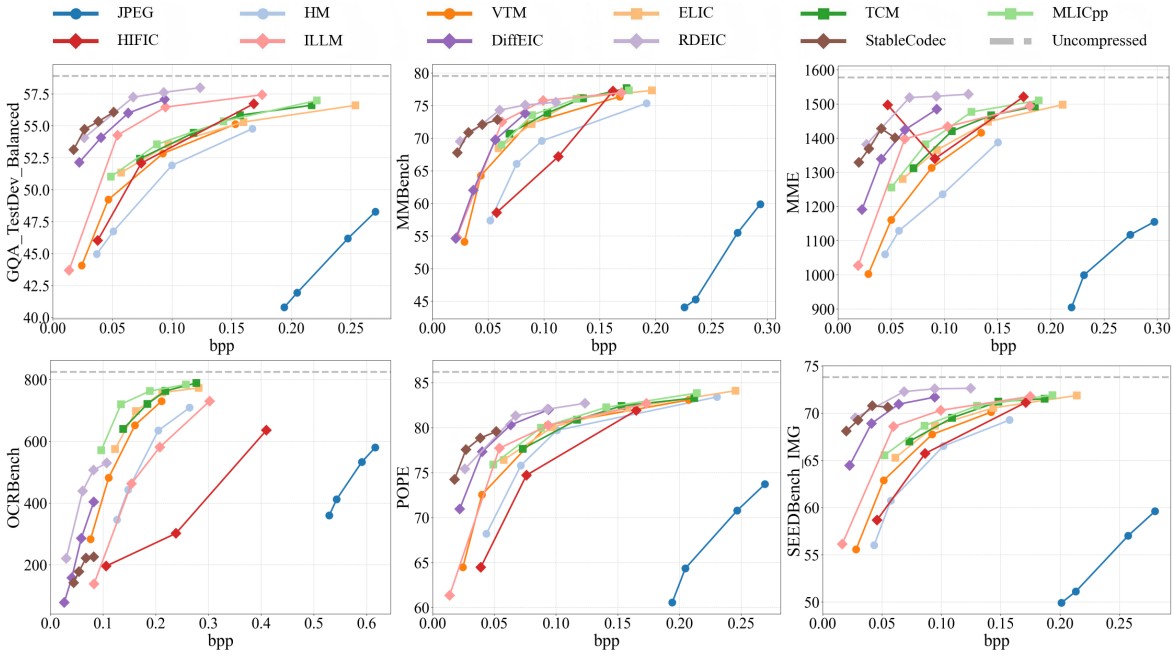

*Figure 3.* Rate-Metric curves for all types of codecs on six tasks using Qwen-VL2.5-3B. Specifically, the GQA metric is computed from five categories of questions. MMBench represents the weighted average of six evaluation dimensions, while MME denotes the aggregate of perception-related measures. OCRBench, POPE, and SEEDBench are weighted averages across five scene types, three sampling strategies, and nine spatial dimensions, respectively.

2024a), MME (Chaoyou et al., 2023), and SEEDBench (SEEDB) (Li et al., 2023a), to evaluate VLMs' performance in perception, semantic understanding, and spatial reasoning with compressed images. All the results are evaluated by VLMEvalKit (Duan et al., 2024) and detailed descriptions of each task and their corresponding evaluation metrics are provided in Appendix B.

**Vision-Language Models.** Since different VLMs exhibit varying task performance, we evaluate several widely used open-source models, including Qwen-Chat-7B (Bai et al., 2023), Qwen-VL2.5-3B, Qwen-VL2.5-7B, Qwen-VL2.5-32B (Bai et al., 2025), InternVL3-1B, InternVL3-2B, InternVL3-8B (Zhu et al., 2025) ,Janus-pro-1B and Janus-pro-7B (Chen et al., 2025), to compare VLMs with different parameter sizes. To quantify the impact of compression, we treat the results on uncompressed images in Appendix Figure 9 as the performance ceiling and measure degradation under different compression settings.

### 3.2. Can VLM Understand Compressed Images

To evaluate whether VLMs can understand heavily images, we follow the setup outlined in Section 3.1 and find that a significant performance gap persists between uncompressed and compressed images for VLM vision. We state our main observations as follows:

*Finding1:* **VLMs exhibit limited understanding of heav-**

**ily compressed images.** Our evaluation indicates that VLMs face significant challenges when dealing with heavily compressed images, as evidenced in Figure 2. All radar plots of the compression methods fall within the boundary of the uncompressed baseline, showing varying degrees of performance degradation. Additionally, we plot the rate-metric curves for all tasks in Figure 3 based on Qwen-VL2.5-3B, and present the rate-metric curves for all other VLMs in Appendix C.1. When the bitrate falls below 0.1 bpp, VLMs struggle to maintain accurate semantic understanding and task performance, as compression artifacts distort crucial visual details. This finding suggests that VLMs' ability to comprehend images deteriorates as compression rates increase, particularly under high compression levels.

*Finding2:* **Stronger VLMs mostly perform better on compressed images.** By comparing the rate–distortion curves across different VLMs and the radar results presented in Figure 9 and Figure 19 in Appendix A and C.2, we observe that their relative performance rankings remain consistent with those under the uncompressed condition. This suggests that models with better performance on uncompressed images also demonstrate greater robustness to compressed images. However, as shown in Figure 2-(a), under the same task conditions, Janus-pro exhibits the smallest decrease in performance across all compression conditions, indicating that it has the best resistance to compression. This resilience is independent of the model's absolute performance.

*Table 2.* Comparison of BD-Metrics for different VLMs across various tasks. The results are computed using each VLM's uncompressed performance and measure the degree of degradation for the corresponding VLM under different codecs. Best in red, second-best in blue.

| VLM | Metric | JPEG | HM | VTM | ELIC | TCM | MLIC | HiFiC | ILLM | DiffEIC | RDEIC | StableCodec |
|---|---|---|---|---|---|---|---|---|---|---|---|---|
| Qwen chat -7B | OCRB | -236.3 | -147.9 | -137.2 | -51.9 | -33.8 | -33.4 | -262.2 | -163.7 | -310.5 | -206.7 | -326.1 |
| | GQA | -8.65 | -3.10 | -3.04 | -1.34 | -1.48 | -1.58 | -2.99 | -2.17 | -1.95 | -1.14 | -2.04 |
| | POPE | -6.36 | -3.01 | -3.46 | -2.09 | -1.90 | -1.65 | -4.05 | -3.12 | -3.05 | -1.71 | -2.66 |
| | MME | -296.9 | -167.7 | -183.6 | -114.5 | -98.4 | -100.8 | -114.9 | -90.8 | -90.4 | -36.1 | -101.9 |
| | MMB | -11.74 | -3.91 | -5.08 | -3.95 | -3.12 | -3.72 | -7.50 | -4.15 | -6.95 | -2.75 | -4.82 |
| | SEEDB | -9.91 | -4.44 | -4.29 | -2.04 | -2.01 | -2.17 | -4.89 | -3.00 | -2.30 | -0.99 | -2.56 |
| | COCOC | -26.79 | -11.12 | -11.53 | -6.78 | -6.01 | -6.74 | -11.05 | -7.90 | -5.78 | -2.45 | -5.76 |
| Intern VL3 -8B | OCRB | -319.9 | -239.7 | -243.6 | -117.1 | -102.1 | -107.5 | -495.6 | -333.5 | -615.4 | -408.5 | -670.1 |
| | GQA | -9.57 | -6.02 | -5.64 | -3.15 | -3.46 | -3.23 | -5.02 | -3.87 | -3.28 | -2.09 | -4.02 |
| | POPE | -7.33 | -3.63 | -3.82 | -2.75 | -2.72 | -2.33 | -6.55 | -4.84 | -5.10 | -2.60 | -4.84 |
| | MME | -266.8 | -172.9 | -151.5 | -95.4 | -89.0 | -91.9 | -231.0 | -132.8 | -173.6 | -92.8 | -175.3 |
| | MMB | -11.69 | -5.83 | -5.74 | -3.34 | -3.41 | -3.48 | -11.89 | -5.74 | -13.48 | -5.87 | -9.59 |
| | SEEDB | -10.06 | -5.72 | -5.55 | -2.80 | -2.62 | -2.88 | -6.38 | -4.44 | -4.00 | -1.78 | -4.22 |
| Qwen VL2.5 -7B | OCRB | -334.6 | -231.8 | -241.9 | -86.5 | -82.3 | -83.8 | -509.9 | -317.3 | -601.4 | -399.6 | -662.1 |
| | GQA | -13.28 | -7.94 | -7.57 | -4.98 | -4.87 | -5.06 | -6.37 | -4.88 | -4.77 | -3.21 | -4.77 |
| | POPE | -18.38 | -9.15 | -9.00 | -6.04 | -6.11 | -6.15 | -11.48 | -8.78 | -8.65 | -6.10 | -8.97 |
| | MME | -522.2 | -295.9 | -311.0 | -162.6 | -165.1 | -169.4 | -231.9 | -161.0 | -184.6 | -93.5 | -220.6 |
| | MMB | -24.71 | -7.88 | -8.17 | -3.57 | -3.76 | -3.77 | -11.80 | -6.49 | -12.58 | -5.66 | -8.35 |
| | SEEDB | -18.00 | -8.94 | -8.04 | -4.01 | -3.85 | -3.83 | -7.82 | -5.16 | -4.29 | -2.17 | -4.49 |
| Janus -pro -7B | OCRB | -259.2 | -175.4 | -186.1 | -84.9 | -86.4 | -85.9 | -312.5 | -205.5 | -380.2 | -246.0 | -413.4 |
| | GQA | -7.16 | -3.61 | -3.28 | -1.84 | -1.80 | -1.50 | -2.81 | -2.42 | -2.17 | -1.03 | -2.46 |
| | POPE | -22.79 | -13.60 | -13.48 | -7.80 | -8.95 | -7.62 | -7.43 | -5.99 | -5.50 | -3.91 | -7.69 |
| | MME | -264.1 | -199.6 | -166.4 | -100.6 | -103.7 | -104.1 | -187.5 | -120.3 | -137.6 | -61.1 | -142.6 |
| | MMB | -10.77 | -5.52 | -4.94 | -2.78 | -2.77 | -3.08 | -9.70 | -5.49 | -9.36 | -3.39 | -7.44 |
| | SEEDB | -7.16 | -4.83 | -4.31 | -1.96 | -2.09 | -2.00 | -4.96 | -3.80 | -2.99 | -1.61 | -3.07 |

*Figure 4.* Rate–Metric curves validating the scaling law of distortion robustness are presented for InternVL3 models (1B, 2B, 8B). Distortion robustness is assessed using OCRBench, POPE, and SEEDBench performance drop relative to the uncompressed results.

*Finding3:* **Generative codecs mostly offer better semantic reconstruction at low bitrates, benefiting VLM-oriented coding.** Compared to traditional and learning-based codecs, generative codecs, particularly those based on diffusion models, are more effective at preserving the semantic content of images. Codecs such as RDEIC and StableCodec excel in reconstructing semantically consistent images at lower bitrates, placing their curves in the upper-left corner of Figure 3, making them more suitable for tasks involving VLMs. In Table 2, we comprehensively evaluate different codecs based on various VLMs and tasks using the ICM task assessment method to measure their BD-Metrics. Additional experimental results can be found in Appendix C.3. The experimental findings further support the notion that generative methods contribute to semantic

understanding (Yan et al., 2025). However, we also observe that generative codecs perform poorly on fine-grained tasks such as OCRBench, which are well known to yield inferior results on text (Xu et al., 2025).

*Finding4:* **The scaling laws for generalization to unseen input don't apply to compressed images.** We conduct scaling law experiments across three codec types and multiple tasks, measuring the performance drop between compressed and uncompressed inputs for models of varying sizes, as shown in Figure 4. Our findings reveal that increasing model size does not consistently reduce compression-induced degradation, indicating a gap between model generalization and robustness to distortion, thus breaking the expected scaling law. Additionally, for StableCodec, the highest bitrate does not yield optimal POPE scores in smaller

models, though this effect diminishes as model size grows. This suggests that VLM generalization capacity modulates the rate-distortion behavior of compression, affecting its monotonicity. Further results are available in Appendix C.4.

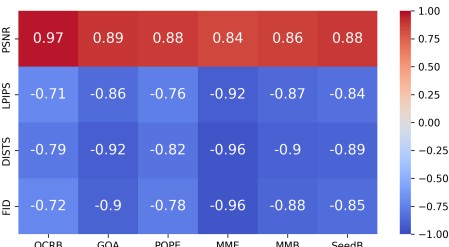

*Figure 5.* Correlation matrix between human-vision metrics and VLM vision tasks. Red represents positive correlation and blue represents negative correlation.

***Finding5:*** **VLM vision tasks correlate with human vision pixel-level metrics, but a gap remains.** To investigate the relationship between human vision and machine perception, we conducted a comparative analysis of VLMs on compressed images using both task-level benchmarks and pixel-level image quality metrics in Figure 5. Specifically, we evaluated VLM performance across multiple downstream tasks while simultaneously measuring image fidelity using perceptual metrics including PSNR, LPIPS, DISTS, and FID. As shown in our results, high pixel-level scores do not always translate to strong VLM task performance, suggesting a decoupling between low-level image quality and semantic understanding. For fine-grained tasks such as OCRBench, pixel-level metrics like PSNR exhibit stronger correlation, while other perceptual metrics show weaker alignment. In contrast, for several coarse-grained tasks, including MMBench, MME, SeedBench, and GQA, DISTS or FID shows the largest absolute correlation coefficient among all the quality metrics. This highlights a nuanced gap between human-centric perceptual metrics and machine vision capabilities, and underscores the need for task-aware evaluation protocols when optimizing image compression for machine vision.

## 4. Understanding the Performance Gap

In this section, we decompose the performance gap between compressed and uncompressed image into two parts: a). The information gap which is caused by the loss of actual information during image compression; b). The generalization gap which is caused by the VLM's failure to generalize compressed images. We discuss the decomposition in detail and visualize those two gaps using numerical examples.

### 4.1. The Information Gap and Generalization Gap

Denote the original image as $X$, the compressed image as $\hat{X}$, the parameter of VLM as $\theta$ and the benchmark performance

as $\mathcal{L}(.,.)$. Then we can decompose the performance gap between compressed image $\hat{X}$ and uncompressed image $X$ into two parts:

$$\underbrace{\mathcal{L}(X,\theta) - \mathcal{L}(\hat{X},\theta)}_{\text{performance gap}} = \underbrace{\mathcal{L}(X,\theta) - \mathcal{L}(\hat{X},\theta^*)}_{\text{information gap}} \quad (1)$$

$$+ \underbrace{\mathcal{L}(\hat{X},\theta^*) - \mathcal{L}(\hat{X},\theta)}_{\text{generalization gap}},$$

$$\text{where } \mathcal{L}(\hat{X},\theta^*) = \max_{\theta} \mathcal{L}(\hat{X},\theta). \quad (2)$$

We name the first part information gap. It is defined as the part of performance gap that can not be resolved no matter how the VLM is finetuned. It is the amount of information about the task that is already lost in image compression process. Given already compressed image $\hat{X}$, there is no remedy to information gap. And the information gap has to be solved by improving the compression algorithm.

We name the second part generalization gap. It is defined as the part of performance gap that is caused by VLM's generalization failure to compressed image. The generalization gap can be reduced by finetuning the VLM using compressed images. In Sec. 5, we propose a VLM adapter to reduce generalization gap for different codecs and bitrates.

*Table 3.* The information and generalization gap on SEEDBench and POPE, with JPEG, ILLM and ELIC as codecs.

| | SEEDBench | | | POPE | | |
|---|---|---|---|---|---|---|
| | JPEG | ELIC | ILLM | JPEG | ELIC | ILLM |
| uncompressed $\mathcal{L}(X,\theta)$ | | 73.81 | | | 86.21 | |
| compressed $\mathcal{L}(\hat{X},\theta)$ | 60.56 | 65.28 | 56.13 | 49.92 | 76.41 | 61.4 |
| compressed finetune $\mathcal{L}(\hat{X},\theta^*)$ | 64.31 | 69.48 | 58.55 | 79.40 | 82.31 | 74.02 |
| Performance gap | 13.25 | 8.53 | 17.68 | 36.29 | 9.8 | 24.81 |
| Information gap | 9.5 | 4.33 | 15.26 | 6.81 | 3.9 | 12.19 |
| Generalization gap | 3.75 | 4.2 | 2.42 | 29.48 | 5.9 | 12.62 |

### 4.2. Visualization of Two Gaps

To better understand the information gap and generalization gap, we provide a numerical example with QwenVL2.5 in Table 3. More specifically, we finetune VLM parameter $\theta$ for JPEG, ELIC and ILLM respectively to enhance the generalization ability of the VLM vision encoder. The reducible gap is generalization gap, and the irreducible gap is information gap. This decomposition should be understood as an empirical framework for distinguishing two practically meaningful sources of performance degradation: the recoverable degradation that can be mitigated by adapting the model to compressed inputs, and the residual degradation that remains unrecovered due to compression-induced information distortion. Since the decomposition depends on the specific adaptation strategy, training configuration, and optimization process, the exact values of these two gaps are generally intractable. Nevertheless, under a given model and

optimization setup, this framework enables us to empirically estimate a lower bound of the Generalization Gap, which correspondingly provides an upper bound of the residual Information Gap. In this sense, the proposed decomposition offers an operational characterization of the two gaps and their empirical bounds, serving as a useful reference for future research on compression-aware VLM adaptation.

## 5. Enhancing with Adapter

### 5.1. Proposed Method

Based on the analysis in Section 4, we propose a unified VLM adapter that can adapt to different types of compression distortions and close the generalization gap. Since fine-tuning the entire VLM requires substantial computational resources, incurs high costs, and is challenging to implement, we found that fine-tuning only the VLM encoder can still yield significant performance gains.

Specifically, existing VLM vision encoders are generally based on the Vision Transformers (ViT) architecture (Han et al., 2022). We need to enable the encoder to understand the distortion types and corresponding compression levels of the compressed images, and incorporate this information as conditional input to the encoder. Assuming there are $m$ existing codecs with different types of distortion, each with $n$ compression levels, we first perform one-hot encoding for each compressor and then map it to the latent variable space through an embedding layer $T(\cdot)$ to get the codec condition embedding $C_{\text{emb}}$, as follows:

$$C_{\text{emb}} = T(m, n, d), \tag{3}$$

where $d$ indicates the embedding dimension aligning with the VLM vision encoder. To inject the codec condition into all spatial positions of the visual tokens, we draw inspiration from the fusion strategy of conditional embeddings and time embeddings in conditional diffusion models (Rombach et al., 2022; Preechakul et al., 2022). We apply rotary positional embedding (RoPE) (Su et al., 2024) to the dimensions $h$ and $w$ obtained after the input image is divided into patches, resulting in an initial positional embedding of dimension $d$ and add the codec condition $C_{\text{emb}}$ to the RoPE representation, obtaining the conditional positional embedding $P_{\text{emb}}$:

$$P_{\text{emb}} = \text{RoPE}(h, w, d) + C_{\text{emb}}. \tag{4}$$

Based on this, we can train the VLM's unified conditional vision encoder (CVE) with parameter $\theta^*$ using compressed images to distill the original vision encoder (VE) with parameter $\theta$, ensuring that the features extracted from uncompressed images $X$ through VE are as similar as possible to the features extracted from compressed images $\hat{X}$ through CVE. To achieve this, we introduce the following distillation

loss $\mathcal{L}_d$, which aims to minimize the mean squared error (MSE) between the two output features:

$$\mathcal{L}_d = \|\text{CVE}(\hat{X}, P_{\text{emb}}, \theta^*) - \text{VE}(X, \theta)\|_2^2. \tag{5}$$

This approach aims to bridge the gap between compressed visual features and the original domain, fine-tuning the VLM to reduce the generalization gap. This distinguishes our method from existing works in the field of coding for machines that focus on fine-tuning for specific codecs.

### 5.2. Experimental Setting

To show our enhancement effect, we utilize QwenVL2.5 as our VLM model and select one representative codec from each of the three compression distortion methods, namely JPEG, ELIC, and ILLM, aligning with Table 3. We used four bitrate levels for training, resulting in a 12-dimensional codec conditional embedding. For training, we use four NVIDIA A100 GPUs in parallel and images are randomly cropped into $336 \times 336$ patches. We train the model with a batch size of 24 for 100k iterations, using an initial learning rate of $1 \times 10^{-4}$. The training dataset consists of over 11w COCO images, compressed at varying bitrates using the three aforementioned codecs. To validate the generalization capability of our proposed method, we conducted two experiments from the codec and VLM perspectives, using different VLMs and previously unseen compressed-distortion images. In addition, we performed comparative and analytical experiments on image coding for machines, which further demonstrate the effectiveness of the information gap and generalization gap.

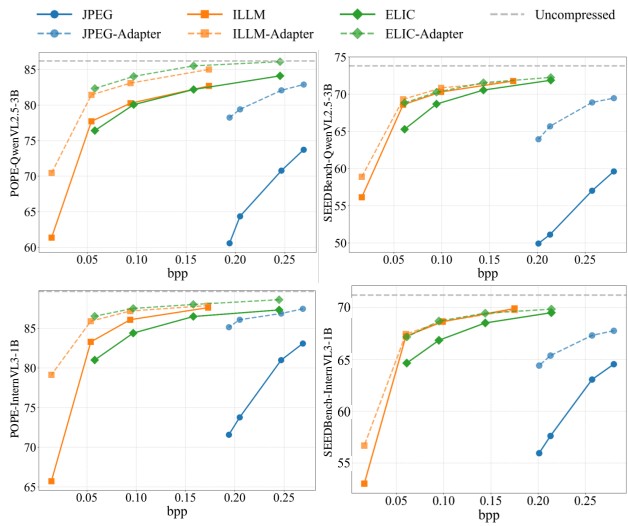

*Figure 6.* Rate-accuracy comparison on POPE and SEEDBench using QwenVL2.5 and InternVL3 across three kinds of codecs.

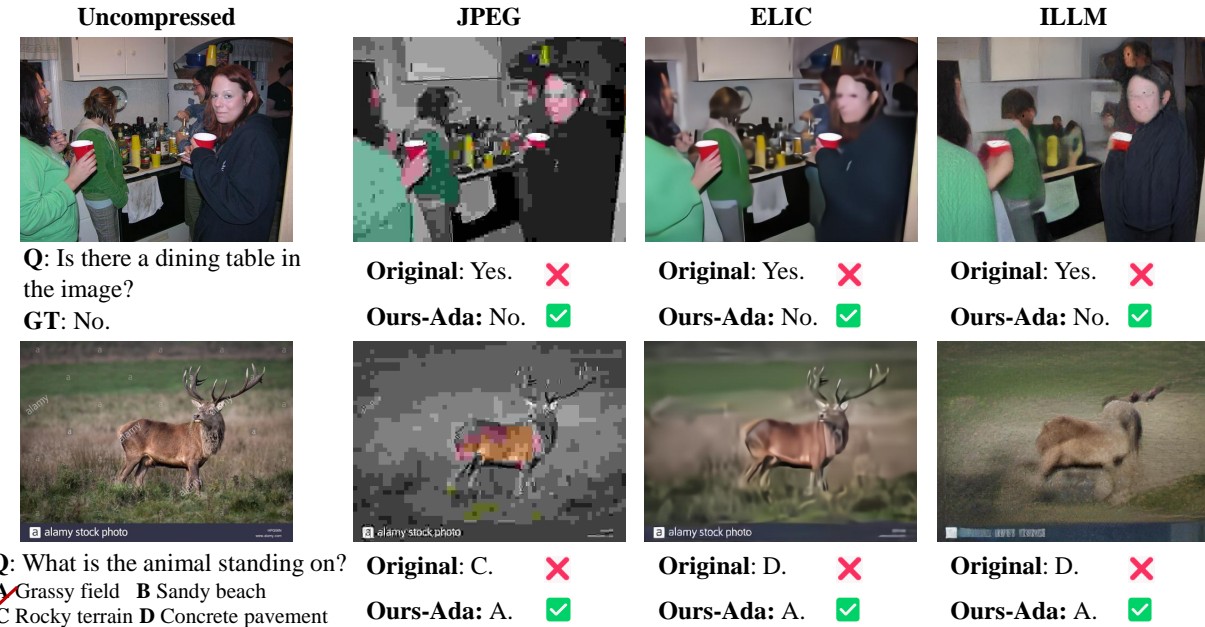

*Figure 7.* Subjective results for POPE and SEEDBench metrics of standard VLM and our method across three kinds of codecs.

### 5.3. Experimental Results

**Quantitative Results.** To quantitatively assess the effectiveness of our adapter-enhanced compression methods, we conduct a rate-accuracy analysis on QwenVL2.5, evaluating its performance across all six benchmarks according to Table 2. Compared to the original VLM model without the adapter, our method achieves a performance gain of over 10 units on all metrics for JPEG at the same bitrate, with also significant improvements on ELIC, ILLM and StableCodec, as shown in Table 4. These results collectively validate the effectiveness of our strategy in preserving semantic fidelity and benchmark performance under aggressive compression. We also provide the rate-accuracy curves in Figure 6 for POPE and SEEDBench, while additional results are provided in Appendix D. Notably, JPEG-Adapter achieves substantial improvements on both metrics, with gains of approximately 30% at low bitrate. Additionally, ELIC and ILLM show over 10% improvement on the POPE metric.

*Table 4.* The BD-Metric results are compared against the original compression outcomes using QwenVL2.5-3B.

| Codec/Metric | POPE | SEEDB | GQA | MMB | OCR | MME |
|---|---|---|---|---|---|---|
| JPEG | 12.62 | 12.88 | 11.63 | 14.91 | 52.51 | 285.4 |
| ELIC | 3.42 | 0.69 | 3.88 | 2.45 | 10.51 | 75.97 |
| ILLM | 3.52 | 1.23 | 2.38 | 0.86 | 14.34 | 19.72 |
| StableCodec | 2.87 | 0.63 | 1.34 | 0.09 | 1.30 | 3.18 |

**Qualitative Results.** To evaluate robustness under compression artifacts, we conducted qualitative comparisons

as shown in Figure 7. Standard VLMs showed significant performance drops, often misinterpreting key visual cues. In contrast, our method consistently produced correct predictions, demonstrating strong resilience to compression-induced distortions. These results suggest that our approach enables reliable multimodal understanding even under heavy lossy compression, making it well-suited for deployment in bandwidth-constrained scenarios.

**Generalization Results.** To test the performance on unseen codec, we evaluate HM, MLICpp, and DiffEIC using our already-trained adapter, treating them as JPEG, ELIC, and MS-ILLM, respectively. The experimental results are shown in Figure 8 and Table 5. From the curves and BD-Metric results, we observe that the adapter yields consistent improvements on all unselected codecs, even though these codecs were never seen during training. Notably, MLICpp achieves larger gains than ELIC itself on SEEDBench, which is both interesting and unexpected. We also notice that DiffEIC improves on POPE, OCRBench and MMB but exhibits a very slight drop on SEEDBench, MME and GQA. This may be attributed to the fact that MS-ILLM is a GAN-based codec, whereas DiffEIC is a diffusion-based codec, leading to a larger semantic and structural gap between the two codec families. Additionally, we validated the effectiveness of our method across different vision encoders, using the InternVL3 model for testing. The experimental results in Figure 6 and across six benchmarks in Table 5 demonstrate consistent performance gains, indicating that our method exhibits strong generalization ability.

**Comparison with ICM.** Our enhancement method shares

*Table 5.* Comparison of BD-Metric results against original compression outcomes using different VLMs and untrained codecs to demonstrate generalization capability.

| VLM | QwenVL2.5-3B | | | InternVL3-1B | | |
|---|---|---|---|---|---|---|
| Codec | HM | MLICpp | DiffEIC | JPEG | ELIC | ILLM |
| POPE | 2.98 | 3.32 | 2.15 | 8.36 | 2.19 | 2.45 |
| SEEDB | 3.12 | 1.22 | -0.16 | 5.62 | 1.19 | 0.42 |
| MME | 130.6 | 50.0 | -3.13 | 133.1 | 25.6 | 7.26 |
| OCRB | 2.10 | 5.73 | 10.51 | 3.93 | 6.75 | 5.64 |
| GQA | 5.48 | 2.01 | -0.37 | 8.58 | 4.17 | 2.24 |
| MMB | 1.25 | 2.52 | 1.20 | 1.40 | 0.86 | 0.73 |

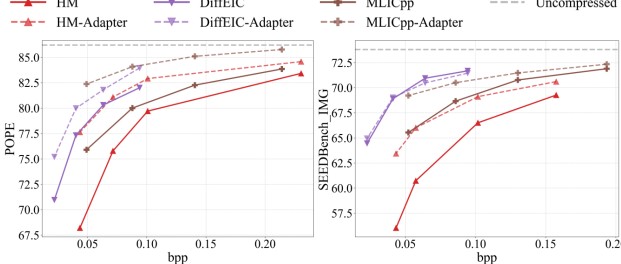

*Figure 8.* Rate-accuracy comparison of QwenVL2.5 on POPE and SEEDBench benchmarks using three untrained codecs.

the same objective as ICM which improves the robustness to compressed images but from different prospective of information and generalization gap. To better illustrate the gap identified in this work, we conducted evaluations of TransTIC (Chen et al., 2023b) on the POPE dataset, as shown in Table 6. First, we tested the results of the base codec TIC (Lu et al., 2022) and fine-tuned the TransTIC with VLM vision encoder to reduce the information gap. We then applied the enhancement method to both models and found that simultaneously addressing both the generalization and information gap yields the best results. A more detailed discussion can be found in the Appendix E.

*Table 6.* The BD-Metric results are compared with ICM methods to visualize the information gap and generalization gap.

| Codec | TIC | TransTIC | TIC-Adapter | TransTIC-Adapter |
|---|---|---|---|---|
| BD-POPE | 0.00 | 0.18 | 2.43 | **3.02** |

### 5.4. Ablation Study

To further investigate the design choices of the proposed adapter, we introduce three additional ablation settings: removing all conditional components, using only distortion-level conditions, and using only codec-identity conditions, as summarized in Table 7. The results show that the adapter still yields consistent performance improvements for VLMs even without any explicit conditional information, suggesting that the proposed adaptation mechanism remains effective in practical blind scenarios where metadata is unavail-

able.

Nevertheless, we observe that the no-condition variant may lead to performance degradation as the bitrate increases. This suggests that, without explicit side information, the adapter may be biased toward learning the correction patterns of more severely distorted low-bitrate samples, thereby impairing its generalization to higher-bitrate images with milder distortions. Introducing the distortion-level condition effectively mitigates this issue and leads to more stable improvements across different compression levels. Furthermore, incorporating codec-identity information brings additional gains, indicating that codec-specific conditioning is beneficial for modeling different artifact patterns and distortion characteristics introduced by different compression methods. In summary, the proposed method, which incorporates both codec-type and distortion-level information, can effectively enhance the performance of VLMs.

*Table 7.* Ablation BD-Metric results under different metadata conditioning settings, including removing all conditional components (*w/o* all meta), using only distortion-level conditions (*w/o* codec), and using only codec-type conditions (*w/o* dist.). Bold font highlights the best values.

| Codec | Metric | *w/o* all | *w/o* codec | *w/o* dist. | Ours |
|---|---|---|---|---|---|
| JPEG | POPE | 11.86 | 12.22 | 12.43 | **12.62** |
| | SEEDB | 11.01 | 11.41 | 12.54 | **12.88** |
| ELIC | POPE | 2.91 | 3.07 | 3.28 | **3.42** |
| | SEEDB | 0.44 | 0.45 | 0.62 | **0.69** |
| ILLM | POPE | 3.16 | 3.19 | 3.41 | **3.52** |
| | SEEDB | 1.14 | 1.18 | 1.21 | **1.23** |

## 6. Conclusion

In this paper, we present a comprehensive benchmark for evaluating VLMs on heavily compressed images, covering over one million samples and assessing both coarse-grained and fine-grained metrics. Our analysis reveals that existing VLMs struggle under compression, and we attribute this to two distinct factors: an inherent information gap due to irreversible data loss, and a generalization gap stemming from poor adaptation to distorted inputs. While the former is difficult to mitigate, we show that the latter can be addressed through model design. To this end, we propose a lightweight VLM adapter that significantly improves performance across diverse codecs and bitrates. Empirical results demonstrate strong gains in recognition accuracy, highlighting the adapter's potential for real-world deployment. We acknowledge that our current experiments do not include the latest proprietary VLM models due to API cost constraints, but we plan to extend our study in future work. We believe this research may contribute to the advancement of image coding for machines and semantic compression.

## Acknowledgement

This work was supported by Wuxi Research Institute of Applied Technologies, Tsinghua University under Grant 20242001120.

## Impact Statement

This paper presents work whose goal is to advance the field of Machine Learning. There are many potential societal consequences of our work, none which we feel must be specifically highlighted here.

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

# A. Detailed Benchmark Setting.

In this section, we provide the detailed codec parameters as shown in Table 8. Our primary focus is to evaluate VLM performance on compressed images under low bitrate conditions; therefore, we selected configurations that yield a bits-per-pixel (bpp) below 0.3. For learning-based and generative codecs, we primarily adopt the pretrained models released by their respective papers. However, for certain codecs such as TCM, MLICpp, and HiFiC, whose default bitrate settings do not align with our target range, we fine-tuned the models using their lowest available bitrate configurations.

Additionally, we report the evaluation results of the selected VLMs on uncompressed images in Figure 9, serving as a reference upper bound for each model-task pair. The metrics are computed using a standardized comparison protocol, which may differ slightly from those reported in the original papers. However, we conducted a careful cross-check and found the discrepancies to be minor. Since our focus is on quantifying the degradation caused by compression, the absolute metric values are less critical than the relative performance drop.

*Table 8.* Three categories of selected image codecs with different bitrate parameters.

| Types | Codecs | P. | Value |
|---|---|---|---|
| Traditional Codecs | JPEG | Q | $\{1, 3, 5, 6\}$ |
| | HM | QP | $\{40\text{-}50\}$ |
| | VTM | QP | $\{40\text{-}53\}$ |
| Learning -based Codecs | ELIC | $\lambda$ | $\{4, 8, 16, 32\}\times 10^{-3}$ |
| | TCM | $\lambda$ | $\{5, 10, 18, 25\}\times 10^{-4}$ |
| | MLICpp | $\lambda$ | $\{4, 9, 18, 35\}\times 10^{-4}$ |
| Generative Codecs | HiFiC | $\lambda$ | $\{2, 6, 8, 14\}\times 10^{-2}$ |
| | MS-ILLM | Q | $\{\text{vlo2}, 1, 2, 3\}$ |
| | DiffEIC | $\lambda$ | $\{2, 4, 8, 16\}$ |
| | RDEIC | $\lambda_r$ | $\{0.1, 0.5, 1, 2\}$ |
| | S.Codec | $\lambda_t$ | $\{2, 4, 8, 16\}$ |

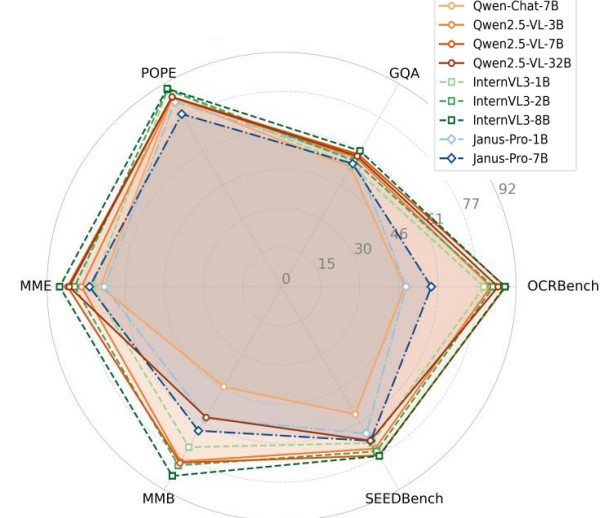

*Figure 9.* Comparison of VLMs on coarse-grained and fine-grained Benchmarks.

# B. Detailed Task Description

We systematically evaluate nine state-of-the-art vision–language models (VLMs) on both compressed and uncompressed images using seven complementary metrics. Table 9 summarizes the evaluation design, including the primary focus of each metric, its sub-metrics, and the number of images considered. Detailed discussions are provided in the following subsections.

In each subsection, we further analyze the impact of image compression by considering four representative codecs. For concreteness, we report detailed per-codec results of Qwen-Chat-7B, providing fine-grained evidence of how compression artifacts affect understanding of VLMs.

*Table 9.* Comparison of seven widely used multimodal evaluation benchmarks: POPE, OCRBench, COCO-Caption, GQA, MMBench, SEED-Bench, and MME.

| Benchmark | Primary Focus | Selected Eval Metrics | Number of Images |
|---|---|---|---|
| POPE | Object hallucination detection | Random,popular and adversarial | 5127 |
| COCO-Caption | Image caption generation | CIDEr, ROUGE$_L$ and BLEU | 5000 |
| OCRBench | Effectiveness in text-related visual tasks | Text Recognition, Scene Text-Centric VQA, Document-Oriented VQA, KIE, and HMER | 1000 |
| GQA | Visual reasoning and compositional question answering | The structural type including choose, compare, logical, query and verify | 398 |
| MME | Perception and cognition abilities | OCR, coarse and fine-grained recognition | 1187 |
| MMBench | Robust and holistic, bilingual VLM evaluation | Six L-2 abilities including AR, CP, FP-C, FP-S, LR and RR | 4329 |
| SEEDBench | Objective evaluation of MLLMs on generative comprehension | 9 dimensions covering image-level and instance-level perception and reasoning | 14232 |

## B.1. POPE (Polling-based Object Probing Evaluation)

We use 5127 images from POPE (Li et al., 2023b), which formulates the evaluation of object hallucination in large VLMs as a binary classification task that prompts LVLMs to output "Yes" or "No". Questions with the answer "No" are built by sampling from negative objects with three different strategies corresponding to the three metrics based-on F1 score reported below, and the column " Overall " is weighted average of these metrics.

- Random: Randomly sample the objects that do not exist in the image.

- Popular: Select the top-k most frequent objects in the whole image dataset that do not exist in the current image, where k is half of the number of polling questions per image.

- Adversarial: First rank all objects according to their co-occurring frequencies with the ground-truth objects, and then select the top-k frequent ones that do not exist in the image.

*Table 10.* Evaluation results of Qwen-Chat-7B on the POPE metric.

| Codec | bpp | Overall | Random | Popular | Adversarial |
|---|---|---|---|---|---|
| JPEG | 0.27 | 82.32 | 79.75 | 83.02 | 84.33 |
|  | 0.25 | 81.09 | 78.28 | 83.25 | 81.91 |
|  | 0.20 | 77.10 | 78.32 | 74.42 | 78.71 |
|  | 0.19 | 75.69 | 73.30 | 77.05 | 76.84 |
| ELIC | 0.25 | 84.77 | 82.48 | 86.85 | 85.10 |
|  | 0.16 | 83.96 | 85.89 | 81.76 | 84.34 |
|  | 0.10 | 83.58 | 83.88 | 85.64 | 81.34 |
|  | 0.06 | 82.10 | 82.39 | 79.82 | 84.21 |
| MSILLM | 0.17 | 85.52 | 83.17 | 87.84 | 85.68 |
|  | 0.09 | 83.96 | 84.10 | 86.38 | 81.53 |
|  | 0.05 | 82.17 | 79.77 | 84.44 | 82.43 |
|  | 0.01 | 73.05 | 72.98 | 70.64 | 75.70 |
| RDEIC | 0.12 | 84.64 | 84.85 | 86.76 | 82.44 |
|  | 0.09 | 85.12 | 87.19 | 82.79 | 85.50 |
|  | 0.07 | 84.46 | 84.98 | 82.18 | 86.33 |
|  | 0.03 | 81.92 | 79.55 | 82.40 | 83.93 |

## B.2. COCO-Caption

COCO-Caption (Chen et al., 2015) provides a standardized dataset and evaluation protocol for image caption generation using 5000 MS COCO testing images with 40 reference sentences per image. Captions output by different approaches are evaluated by automatic metrics including CIDEr, ROUGE$_L$ and Bleu.

- CIDEr aggregates Term Frequency Inverse Document Frequency (TF–IDF) weighted n-gram cosine similarity.

- ROUGE$_L$ computes an F-measure based on the longest common subsequence (LCS) between candidate and reference captions.

- BLEU analyzes the co-occurrences of n-grams between the candidate and reference sentences and computes a corpus-level clipped n-gram precision with a brevity penalty:

$$\text{BLEUK} = \text{BP} \cdot \exp\Big(\sum_{k=1}^{K} w_k \log p_k\Big),$$

where $p_k$ are clipped $k$-gram precisions and BP is the brevity penalty.

*Table 11.* Evaluation results of Qwen-Chat-7B on the COCO-Caption metric.

| Codec | bpp | CIDEr | ROUGE$_L$ | Bleu1 | Bleu2 | Bleu3 | Bleu4 |
|---|---|---|---|---|---|---|---|
| JPEG | 0.27 | 83.09 | 51.08 | 71.82 | 54.20 | 39.97 | 29.36 |
| | 0.25 | 78.37 | 50.32 | 70.94 | 52.76 | 38.51 | 28.09 |
| | 0.20 | 60.35 | 45.94 | 65.12 | 45.91 | 32.06 | 22.49 |
| | 0.19 | 56.37 | 44.97 | 63.69 | 44.31 | 30.60 | 21.18 |
| ELIC | 0.26 | 95.73 | 54.06 | 75.09 | 58.01 | 43.76 | 32.79 |
| | 0.16 | 93.02 | 53.67 | 74.49 | 57.40 | 42.97 | 31.96 |
| | 0.10 | 88.33 | 52.62 | 73.46 | 55.85 | 41.48 | 30.75 |
| | 0.06 | 81.56 | 50.84 | 71.53 | 53.49 | 39.17 | 28.64 |
| MSILLM | 0.17 | 96.66 | 54.45 | 75.37 | 58.47 | 44.34 | 33.43 |
| | 0.09 | 95.20 | 54.10 | 75.25 | 58.32 | 44.02 | 33.00 |
| | 0.05 | 90.84 | 53.35 | 74.21 | 57.00 | 42.66 | 31.77 |
| | 0.01 | 53.11 | 44.49 | 62.79 | 43.16 | 29.68 | 20.61 |
| RDEIC | 0.12 | 97.84 | 54.66 | 75.74 | 58.92 | 44.78 | 33.76 |
| | 0.09 | 97.05 | 54.37 | 75.55 | 58.78 | 44.57 | 33.56 |
| | 0.07 | 97.00 | 54.69 | 75.70 | 58.91 | 44.69 | 33.62 |
| | 0.03 | 89.30 | 53.00 | 73.65 | 56.34 | 42.16 | 31.43 |

## B.3. OCRBench

We use 1000 images from OCRBench (Liu et al., 2024b), a comprehensive benchmark assessing Optical Character Recognition (OCR) capabilities in LLMs through five text-related visual tasks including Text Recognition, Scene Text-Centric Visual Question Answering (VQA), Document-Oriented VQA, Key Information Extraction (KIE), and Handwritten Mathematical Expression Recognition (HMER).

- Text Recognition: Evaluate LMM with widely-adopted OCR text recognition datasets from 8 perspectives.

- Scene Text-Centric VQA: Test LLMs on five datasets.

- Document-Oriented VQA: Assess LLMs on three datasets.

- Key Information Extraction: Conduct experiments on three datasets.

- Handwritten Mathematical Expression Recognition: Evaluate on HME100K.

In the table below, the above metrics are abbreviated for "Text Reco", "Scene VQA", "Doc VQA", "KIE", "HMER" respectively and the total sum is "Final Score".

## B.4. GQA

We use 398 image from GQA (Hudson & Manning, 2019), a large-scale dataset for real-world visual reasoning and compositional question answering. We associate each question with the structural type derived from the final operation in the question's functional program, as shown below, with the "Overall" stands for the weighted sum.

- choose: Questions that present two alternatives to choose from, e.g. "Is it red or blue ?"

- compare: Comparison questions between two or more objects.

- logical: Involve logical inference.

- query: All open questions.

- verify: Yes/No questions.

*Table 12.* Evaluation results of Qwen-Chat-7B on the OCRBench metric.

| Codec | bpp | Final Score | Text Reco | Scene VQA | Doc VQA | KIE | HMER |
|---|---|---|---|---|---|---|---|
| JPEG | 0.62 | 311.00 | 68.00 | 119.00 | 70.00 | 54.00 | 0.00 |
| | 0.59 | 280.00 | 57.00 | 108.00 | 62.00 | 53.00 | 0.00 |
| | 0.54 | 202.00 | 39.00 | 85.00 | 45.00 | 33.00 | 0.00 |
| | 0.53 | 173.00 | 31.00 | 78.00 | 37.00 | 27.00 | 0.00 |
| ELIC | 0.28 | 467.00 | 167.00 | 145.00 | 85.00 | 70.00 | 0.00 |
| | 0.21 | 453.00 | 163.00 | 140.00 | 84.00 | 66.00 | 0.00 |
| | 0.16 | 416.00 | 154.00 | 125.00 | 80.00 | 57.00 | 0.00 |
| | 0.12 | 336.00 | 138.00 | 106.00 | 49.00 | 43.00 | 0.00 |
| MSILLM | 0.30 | 435.00 | 162.00 | 133.00 | 81.00 | 59.00 | 0.00 |
| | 0.21 | 366.00 | 145.00 | 115.00 | 66.00 | 40.00 | 0.00 |
| | 0.15 | 289.00 | 128.00 | 94.00 | 37.00 | 30.00 | 0.00 |
| | 0.08 | 101.00 | 49.00 | 34.00 | 18.00 | 0.00 | 0.00 |
| RDEIC | 0.11 | 327.00 | 151.00 | 112.00 | 40.00 | 24.00 | 0.00 |
| | 0.08 | 310.00 | 138.00 | 104.00 | 43.00 | 25.00 | 0.00 |
| | 0.06 | 284.00 | 129.00 | 100.00 | 34.00 | 21.00 | 0.00 |
| | 0.03 | 166.00 | 80.00 | 61.00 | 18.00 | 7.00 | 0.00 |

## B.5. MME (Comprehensive Multimodal LLM (MLLM) Evaluation)

MME (Chaoyou et al., 2023) aims to offer a comprehensive evaluation suite that jointly measures perception and cognition for MLLMs across 14 subtasks. We only test the perception part for 1187 compressed images, which on top of OCR includes the recognition of coarse-grained and fine-grained objects. The "Overall" stands for the sum of all perception metrics.

- OCR: Optical Character Recognition (OCR) is a foundational capability of MLLMs, serving for subsequent text-based tasks such as text translation and text understanding.

- Coarse-Grained Recognition: The contents of coarse- grained recognition include the existence of common objects, and their count, color, and position. In each perception subtask, we prepare 30 images with 60 instruction-answer pairs.

- Fine-Grained Recognition: The fine-grained recog- nition is more about testing the knowledge resources of MLLMs. The subtasks consist of recognizing movie posters, celebrities, scenes, landmarks, and artworks, containing 147, 170, 200, 200, and 200 images respectively.

## B.6. MMBench

MMBench (Liu et al., 2024a) is a systematically designed objective multi-modality benchmark for a robust and holistic evaluation of VLMs with 4329 images covering 20 ability dimensions. "Overall" is the weighted average of the above six metrics.

- AR: Attribute Reasoning.

- CP: Coarse Perception

- Fine-grained Perception: FP-C, cross-instance; FP-S, single-instance.

- LR: Logical Reasoning.

- RR: Relation Reasoning.

*Table 13.* Evaluation results of Qwen-Chat-7B on the GQA metric.

| Codec | bpp | Overall | choose | compare | logical | query | verify |
|---|---|---|---|---|---|---|---|
| JPEG | 0.27 | 48.18 | 70.68 | 55.18 | 63.12 | 31.21 | 74.38 |
| | 0.25 | 47.42 | 70.68 | 53.14 | 61.62 | 30.46 | 74.11 |
| | 0.20 | 43.20 | 64.92 | 52.63 | 58.74 | 26.48 | 67.94 |
| | 0.19 | 41.93 | 63.95 | 53.99 | 57.24 | 24.89 | 66.96 |
| ELIC | 0.25 | 54.06 | 75.82 | 57.22 | 69.22 | 37.68 | 79.71 |
| | 0.16 | 53.12 | 75.29 | 55.35 | 68.94 | 36.65 | 78.51 |
| | 0.10 | 52.67 | 73.16 | 55.52 | 69.72 | 36.11 | 78.06 |
| | 0.06 | 52.31 | 74.22 | 55.86 | 65.56 | 36.40 | 77.89 |
| MSILLM | 0.18 | 53.91 | 76.26 | 55.69 | 67.05 | 37.97 | 79.88 |
| | 0.09 | 53.32 | 75.38 | 55.69 | 66.72 | 37.49 | 78.73 |
| | 0.05 | 52.31 | 74.22 | 55.86 | 65.56 | 36.40 | 77.89 |
| | 0.01 | 43.71 | 66.52 | 52.97 | 61.90 | 26.42 | 67.54 |
| RDEIC | 0.12 | 53.83 | 75.91 | 56.54 | 66.06 | 37.90 | 80.42 |
| | 0.09 | 53.69 | 76.53 | 56.88 | 65.45 | 37.91 | 79.66 |
| | 0.07 | 53.54 | 76.53 | 55.86 | 65.95 | 37.74 | 79.22 |
| | 0.03 | 51.69 | 74.49 | 53.99 | 66.39 | 35.24 | 77.58 |

## B.7. SEEDBench

SEED-Bench (Li et al., 2023a) consists of 19K multiple choice questions with accurate human annotations, which spans 12 evaluation dimensions including the comprehension of both the image and video modality. We evaluate VLMs on 14232 images across 9 dimensions, involving only spatial understanding. "Overall" is the weighted average of the above metrics.

- Instance Attributes: This dimension is related to the attributes of an instance, such as color, shape or material. It assesses a model's understanding of an object's visual appearance.

- Instance Identity: This dimension involves the identification of a certain instance in the image, including the existence or category of a certain object in the image. It evaluates a model's object recognition capability.

- Instance Interaction: This dimension requires the model to recognize the state relation or interaction relations between two humans or objects.

- Instance Location: This dimension concerns the absolute position of one specified instance. It requires a model to correctly localize the object referred to in the question.

- Instances Counting: This dimension requires the model to count the number of a specific object in the image. This requires the model to understand all objects, and successfully count the referred object's instances.

- Scene Understanding: This dimension focuses on the global information in the image. Questions can be answered through a holistic understanding of the image.

- Spatial Relation: This dimension asks an model to ground the two mentioned objects, and recognize their relative spatial relation within the image.

- Text Understanding: For this dimension, the model should answer question about the textual elements in the image.

- Visual Reasoning: This dimension evaluates if a model is able to reason based on the visual information. This requires the model to fully understand the image and utilize its common sense knowledge to correctly answer the questions.

In the table below, the above metrics are abbreviated for "Attr.", "Ident." , "Interact.", "Loc.", "Count", "Scene", "Spatial", "Text", "Reason." respectively.

*Table 14.* Evaluation results of Qwen-Chat-7B on the MME metric.

| Codec | bpp | Overall | OCR | artwork | celebrity | color | count | existence | landmark | position | posters | scene |
|-------|-----|---------|-----|---------|-----------|-------|-------|-----------|----------|----------|---------|-------|
| JPEG | 0.30 | 1236.7 | 80.0 | 95.0 | 76.8 | 146.7 | 108.3 | 180.0 | 108.0 | 128.3 | 150.3 | 163.2 |
| | 0.27 | 1208.5 | 87.5 | 87.8 | 73.5 | 141.7 | 103.3 | 180.0 | 97.0 | 120.0 | 153.7 | 164.0 |
| | 0.23 | 1017.5 | 72.5 | 74.0 | 53.5 | 128.3 | 93.3 | 151.7 | 74.2 | 81.7 | 135.7 | 152.5 |
| | 0.22 | 971.8 | 62.5 | 75.5 | 52.4 | 138.3 | 76.7 | 140.0 | 66.5 | 86.7 | 127.6 | 145.8 |
| ELIC | 0.21 | 1367.6 | 65.0 | 103.0 | 109.4 | 175.0 | 138.3 | 185.0 | 141.0 | 128.3 | 156.8 | 165.8 |
| | 0.14 | 1321.7 | 80.0 | 96.2 | 100.0 | 163.3 | 123.3 | 180.0 | 136.2 | 121.7 | 155.1 | 165.8 |
| | 0.09 | 1288.4 | 72.5 | 95.0 | 85.6 | 170.0 | 126.7 | 175.0 | 123.5 | 121.7 | 151.0 | 167.5 |
| | 0.06 | 1226.8 | 72.5 | 90.2 | 71.2 | 161.7 | 106.7 | 175.0 | 114.5 | 123.3 | 149.0 | 162.8 |
| MSILLM | 0.18 | 1382.6 | 72.5 | 112.0 | 115.9 | 180.0 | 125.0 | 185.0 | 147.5 | 116.7 | 163.3 | 164.8 |
| | 0.10 | 1394.9 | 87.5 | 115.5 | 113.5 | 185.0 | 123.3 | 185.0 | 139.2 | 126.7 | 156.1 | 163.0 |
| | 0.06 | 1325.6 | 65.0 | 110.2 | 93.2 | 175.0 | 106.7 | 185.0 | 136.0 | 135.0 | 153.4 | 166.0 |
| | 0.02 | 1066.4 | 80.0 | 95.0 | 40.6 | 148.3 | 105.0 | 140.0 | 98.0 | 110.0 | 102.7 | 146.8 |
| RDEIC | 0.12 | 1408.4 | 72.5 | 119.2 | 120.9 | 180.0 | 140.0 | 190.0 | 146.5 | 125.0 | 151.0 | 163.2 |
| | 0.09 | 1428.6 | 95.0 | 122.2 | 118.8 | 175.0 | 140.0 | 190.0 | 147.8 | 130.0 | 147.3 | 162.5 |
| | 0.07 | 1395.9 | 87.5 | 119.0 | 107.4 | 170.0 | 135.0 | 195.0 | 146.2 | 125.0 | 147.3 | 163.5 |
| | 0.03 | 1311.4 | 95.0 | 111.2 | 84.1 | 165.0 | 110.0 | 185.0 | 141.8 | 115.0 | 139.8 | 164.5 |

*Table 15.* Evaluation results of Qwen-Chat-7B on the MMBench metric.

| Codec | bpp | Overall | AR | CP | FP-C | FP-S | LR | RR |
|-------|-----|---------|-----|-----|------|------|-----|-----|
| JPEG | 0.29 | 36.68 | 35.68 | 48.99 | 30.77 | 44.71 | 12.71 | 18.26 |
| | 0.27 | 35.91 | 36.18 | 45.61 | 30.77 | 45.39 | 13.56 | 15.65 |
| | 0.24 | 30.67 | 31.16 | 41.89 | 25.87 | 35.84 | 14.41 | 10.43 |
| | 0.23 | 27.66 | 33.17 | 35.81 | 20.98 | 30.38 | 16.10 | 10.43 |
| ELIC | 0.20 | 43.56 | 38.19 | 60.47 | 37.06 | 51.19 | 16.95 | 25.22 |
| | 0.13 | 41.75 | 36.18 | 60.14 | 37.06 | 48.46 | 14.41 | 20.87 |
| | 0.09 | 41.49 | 37.19 | 58.11 | 34.97 | 49.83 | 16.10 | 19.13 |
| | 0.06 | 37.63 | 35.18 | 51.69 | 31.47 | 44.03 | 12.71 | 22.61 |
| MSILLM | 0.17 | 42.53 | 38.69 | 61.15 | 30.77 | 49.83 | 14.41 | 26.09 |
| | 0.10 | 43.64 | 39.20 | 60.47 | 35.66 | 51.19 | 16.10 | 26.96 |
| | 0.06 | 42.70 | 41.21 | 60.14 | 37.06 | 49.49 | 10.17 | 23.48 |
| | 0.02 | 26.80 | 25.63 | 40.20 | 23.78 | 29.69 | 11.02 | 6.96 |
| RDEIC | 0.11 | 42.96 | 37.69 | 59.80 | 37.06 | 50.51 | 15.25 | 25.22 |
| | 0.08 | 43.56 | 35.68 | 60.14 | 37.76 | 52.22 | 15.25 | 28.70 |
| | 0.06 | 43.21 | 36.68 | 59.46 | 37.06 | 52.22 | 15.25 | 26.09 |
| | 0.02 | 40.55 | 35.18 | 59.12 | 32.17 | 46.76 | 12.71 | 25.22 |

*Table 16.* Evaluation results of Qwen-Chat-7B on the SEEDBench metric.

| Codec | bpp | Overall | Attr. | Ident. | Interact. | Loc. | Count | Scene | Spatial | Text | Reason. |
|-------|-----|---------|-------|--------|-----------|------|-------|-------|---------|------|---------|
| JPEG | 0.28 | 50.71 | 51.09 | 56.47 | 63.92 | 47.03 | 35.39 | 61.94 | 41.86 | 20.24 | 51.96 |
|  | 0.26 | 49.47 | 50.16 | 53.96 | 57.73 | 46.11 | 35.31 | 60.29 | 39.88 | 16.67 | 51.06 |
|  | 0.21 | 46.11 | 46.72 | 48.17 | 52.58 | 43.35 | 34.16 | 56.68 | 35.31 | 22.62 | 47.13 |
|  | 0.20 | 44.23 | 45.30 | 45.82 | 46.39 | 39.47 | 32.16 | 54.08 | 36.99 | 22.62 | 48.94 |
| ELIC | 0.21 | 56.74 | 57.90 | 63.35 | 55.67 | 54.29 | 43.11 | 65.83 | 42.31 | 29.76 | 60.73 |
|  | 0.14 | 56.42 | 58.57 | 62.42 | 60.82 | 52.35 | 42.01 | 65.67 | 41.70 | 25.00 | 58.91 |
|  | 0.09 | 55.09 | 56.72 | 61.28 | 58.76 | 51.23 | 40.25 | 64.88 | 42.62 | 28.57 | 56.19 |
|  | 0.06 | 55.51 | 57.65 | 60.13 | 57.73 | 51.74 | 41.07 | 65.55 | 41.55 | 30.95 | 55.29 |
| MSILLM | 0.17 | 57.35 | 59.09 | 63.41 | 57.73 | 53.58 | 43.69 | 66.34 | 43.07 | 26.19 | 61.93 |
|  | 0.10 | 56.51 | 58.06 | 61.66 | 59.79 | 53.27 | 42.30 | 66.37 | 43.53 | 26.19 | 59.21 |
|  | 0.06 | 55.51 | 57.65 | 60.13 | 57.73 | 51.74 | 41.07 | 65.55 | 41.55 | 30.95 | 55.29 |
|  | 0.02 | 43.25 | 46.29 | 44.40 | 51.55 | 39.98 | 31.30 | 51.49 | 31.66 | 22.62 | 39.27 |
| RDEIC | 0.12 | 57.95 | 59.71 | 64.12 | 58.76 | 54.70 | 44.54 | 66.50 | 44.44 | 25.00 | 61.33 |
|  | 0.09 | 57.51 | 59.17 | 63.68 | 56.70 | 54.09 | 44.14 | 66.18 | 44.44 | 23.81 | 61.03 |
|  | 0.07 | 57.37 | 58.34 | 63.52 | 61.86 | 54.91 | 44.42 | 66.50 | 44.29 | 22.62 | 59.21 |
|  | 0.03 | 54.79 | 56.74 | 60.19 | 59.79 | 51.43 | 40.95 | 63.77 | 41.86 | 26.19 | 55.59 |

## C. More Benchmarking Results

### C.1. Additional Rate-Metric curves Results

In this work, we primarily evaluate open-source VLMs because the cost of querying commercial API-based models is prohibitively high, as stated in the conclusion section of the original paper. Moreover, closed-source models cannot be used for our adapter-based enhancement experiments, making it impossible to verify the effectiveness of the proposed method on these models. However, we conducted an evaluation on OCRBench using GPT-4o to validate the effectiveness for closed-source VLMs. The rate-metric curve is shown in Figure 10 and all outcomes are fully consistent with the findings presented in this paper.

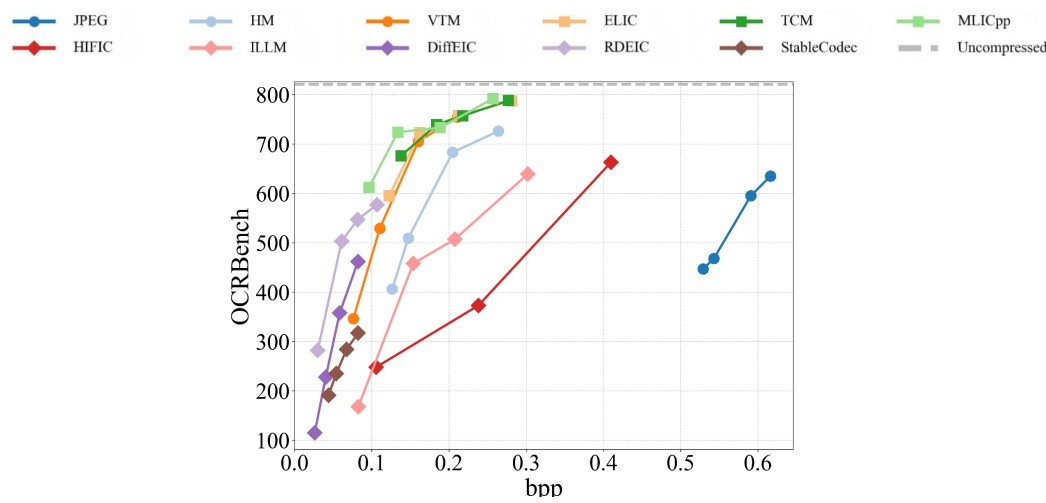

*Figure 10.* Rate-OCRBench curve for all types of codecs using Chatgpt-4o.

We further incorporated the COCO-Caption task using the Qwen-Chat model as shown in Figure 11. By including this image captioning task, we enhance the breadth and completeness of our evaluation. Representative experimental results are presented below. We report the bpp, ROUGE-L, and CIDEr scores for three representative methods, and use CIDEr to compute the BD-Metric for all compression approaches. The outcomes remain fully consistent with the findings reported in our paper. From the curve, we observe that generative codecs achieve the best overall performance, further confirming their advantage in semantic reconstruction under low-bitrate conditions.

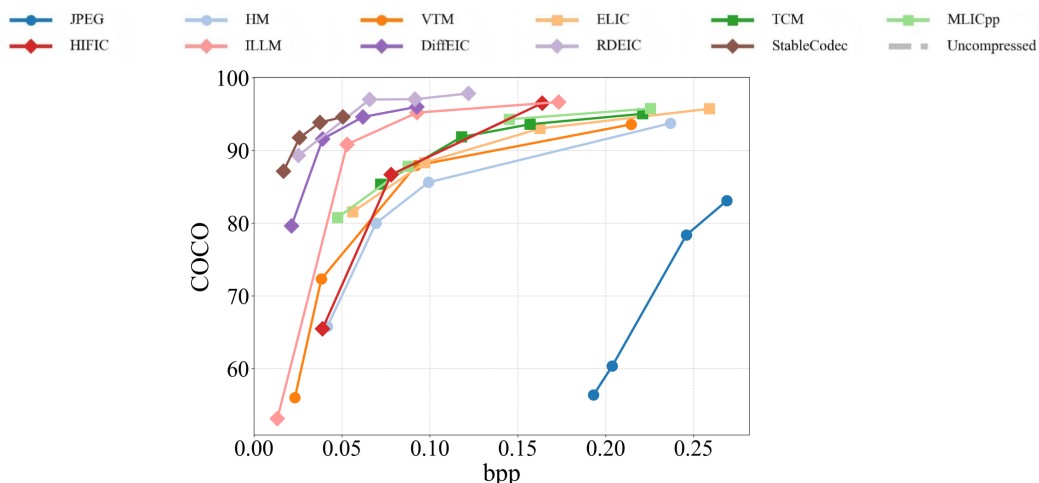

*Figure 11.* Rate-metric curve based on COCO-Caption task for all types of codecs using Qwen-Chat model.

Furthermore, we present additional evaluation results in the form of Rate–Metric curves in Figure 12, Figure 13, Figure 14, Figure 15, Figure 16, Figure 17 and Figure 18. We report results for other VLMs, organized into three series. Each figure

includes Rate–Metric curves across all considered codecs and benchmark datasets as well as the baseline performance on uncompressed images.

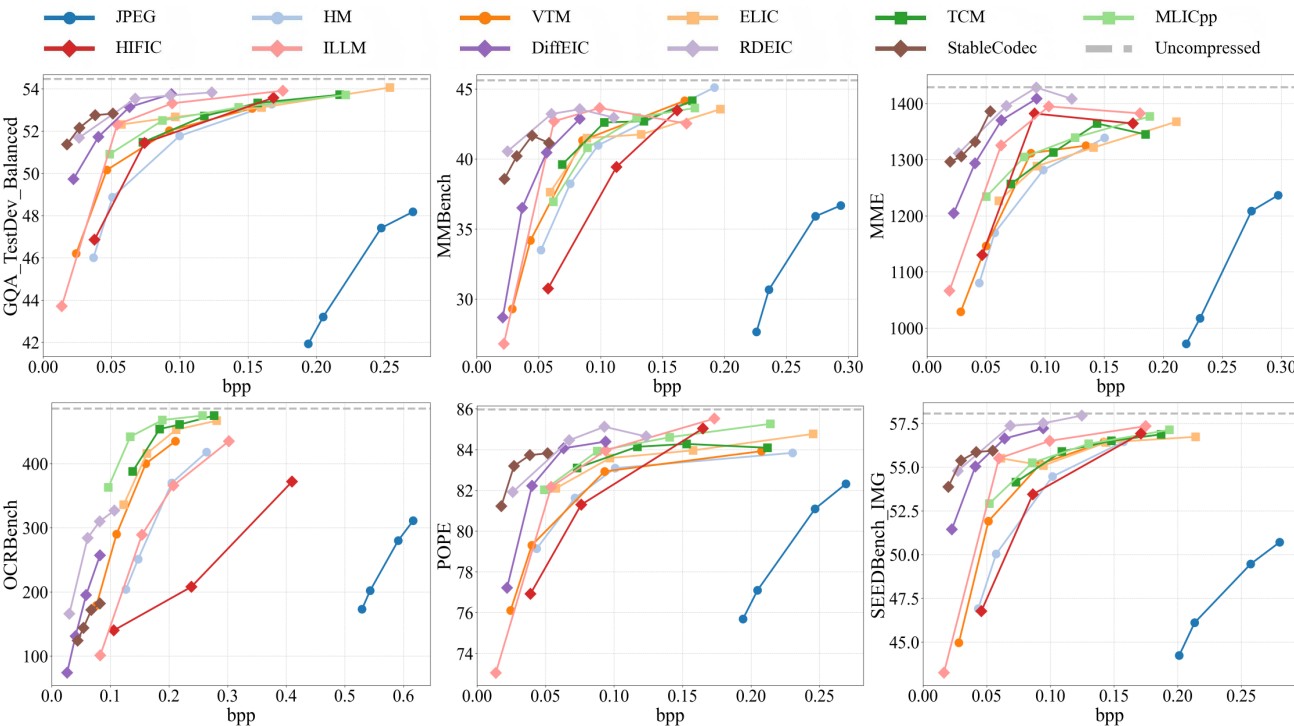

*Figure 12.* Rate-Metric curves for all types of codecs on GQA, MMB, MME, OCRBench, POPE, and SEEDBench using Qwen-Chat-7B.

## C.2. Additional VLMs Compression Results

Figure 19 presents radar charts comparing the performance of various VLMs under three different compression codecs: JPEG, ELIC, and ILLM. Each chart visualizes model performance across six benchmarks: POPE, GQA, SEEDBench, MMB, MME, and OCRBench.Under comparable parameter scales,InternVL3 consistently outperforms Qwen2.5, which in turn surpasses Janus-Pro across all distortion conditions. This ranking is consistent with the uncompressed baseline results reported in Figure 9, reinforcing the observation that stronger models exhibit greater robustness to compression artifacts.

## C.3. Additional BD-Metrics Results

We report additional evaluation results for VLMs, which also serve as the data source for Figure 2.

## C.4. Additional Scaling Law Results

In this section, we provide additional metrics for the InternVL3 model series, as shown in Figure 20. These results are consistent with the main findings: the scaling laws don't apply to compressed images.

Additionally, we report results for the Qwen2.5-VL model series with 3B, 7B, and 32B parameters across different codecs, as shown in Figure 21. Interestingly, the 32B variant underperforms the 7B model on uncompressed images. Due to the lack of publicly available technical details for Qwen2.5-VL-32B, we are unable to verify this discrepancy. Nevertheless, the overall trend is clear: larger model size does not necessarily correspond to lower performance drop under compression.

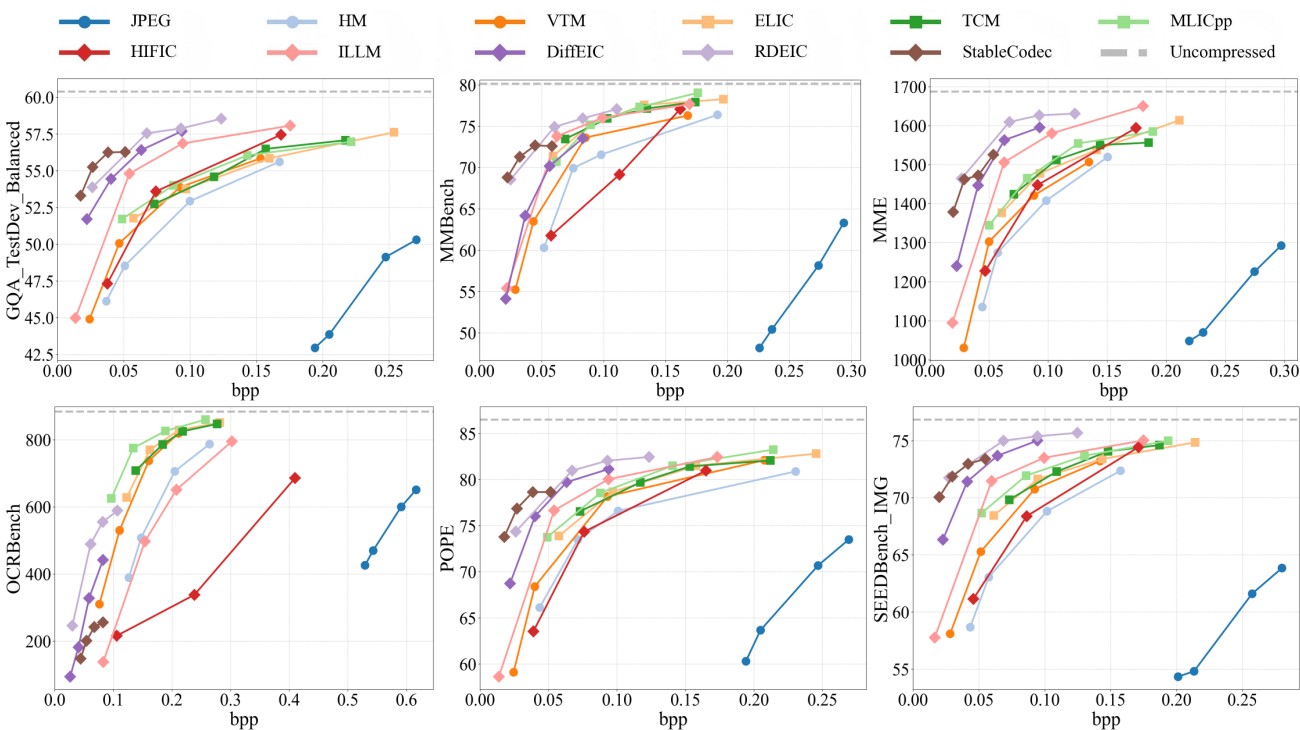

*Figure 13.* Rate-Metric curves for all types of codecs on GQA, MMB, MME, OCRBench, POPE, and SEEDBench using Qwen2.5-VL-7B.

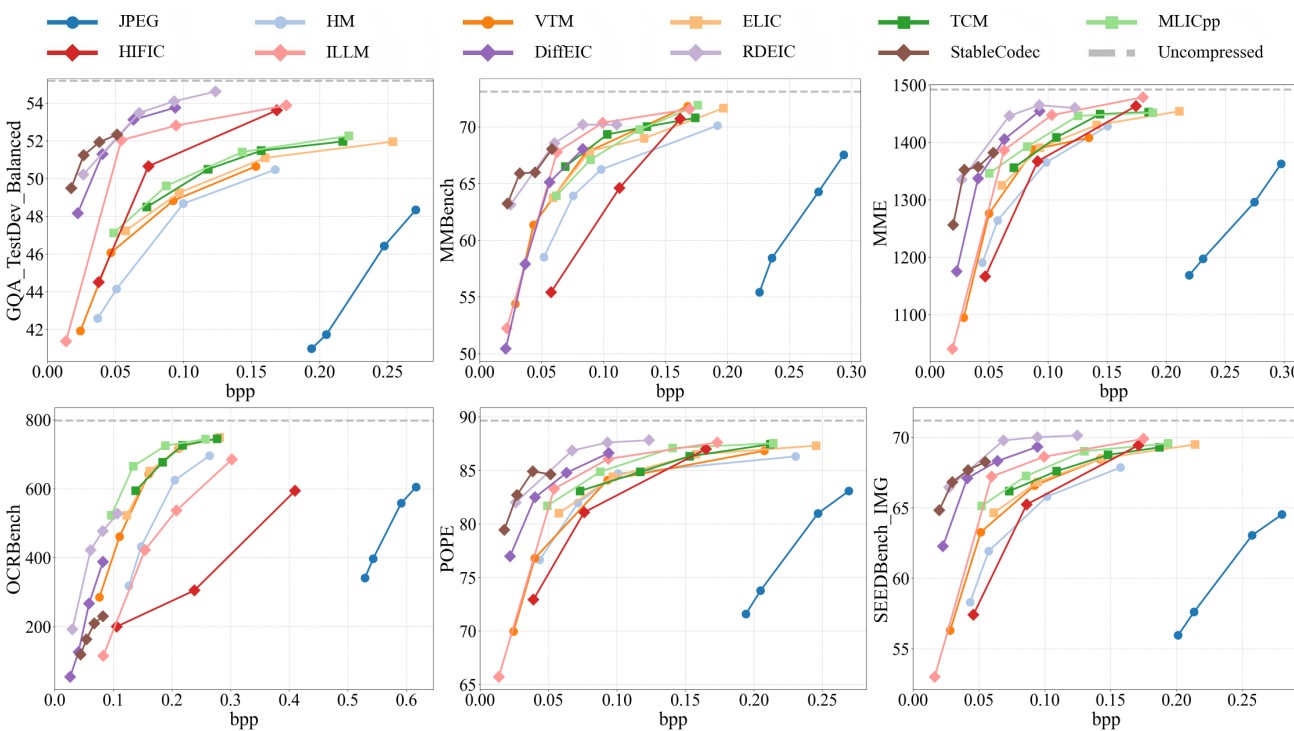

*Figure 14.* Rate-Metric curves for all types of codecs on GQA, MMB, MME, OCRBench, POPE, and SEEDBench using InternVL3-1B.

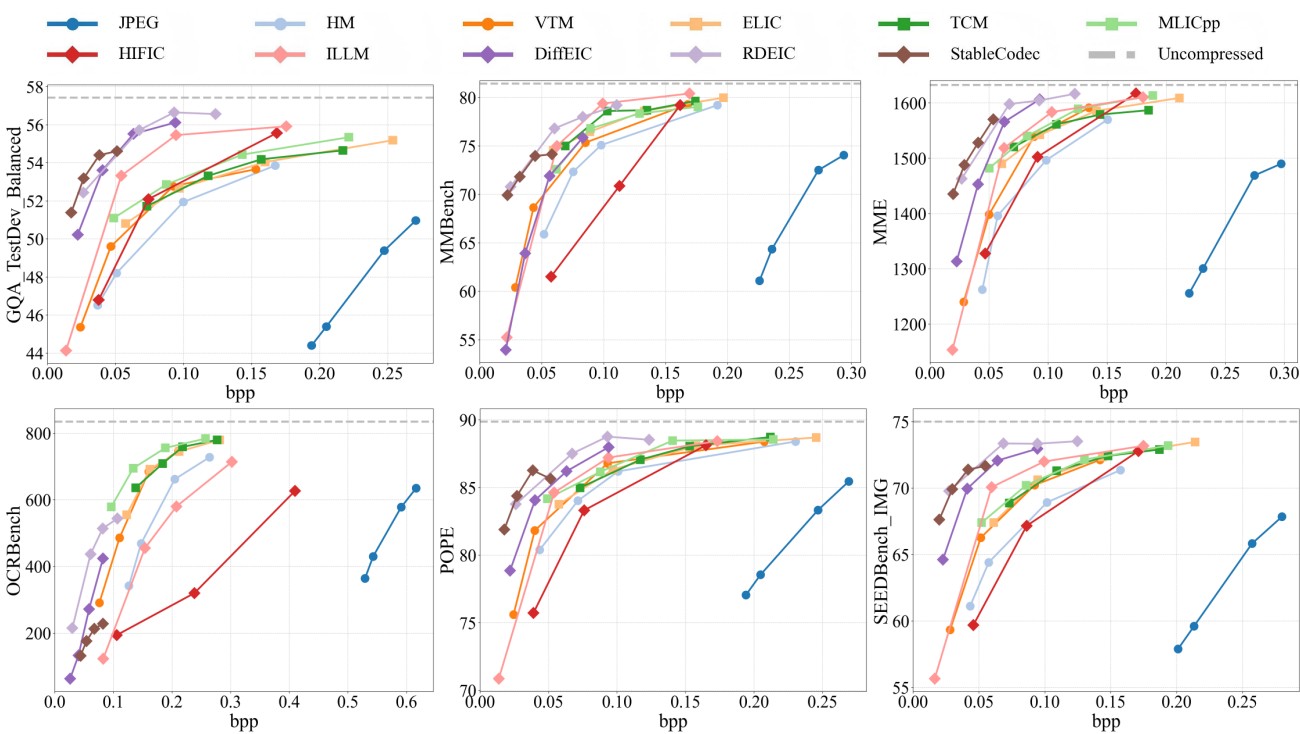

*Figure 15.* Rate-Metric curves for all types of codecs on GQA, MMB, MME, OCRBench, POPE, and SEEDBench using InternVL3-2B.

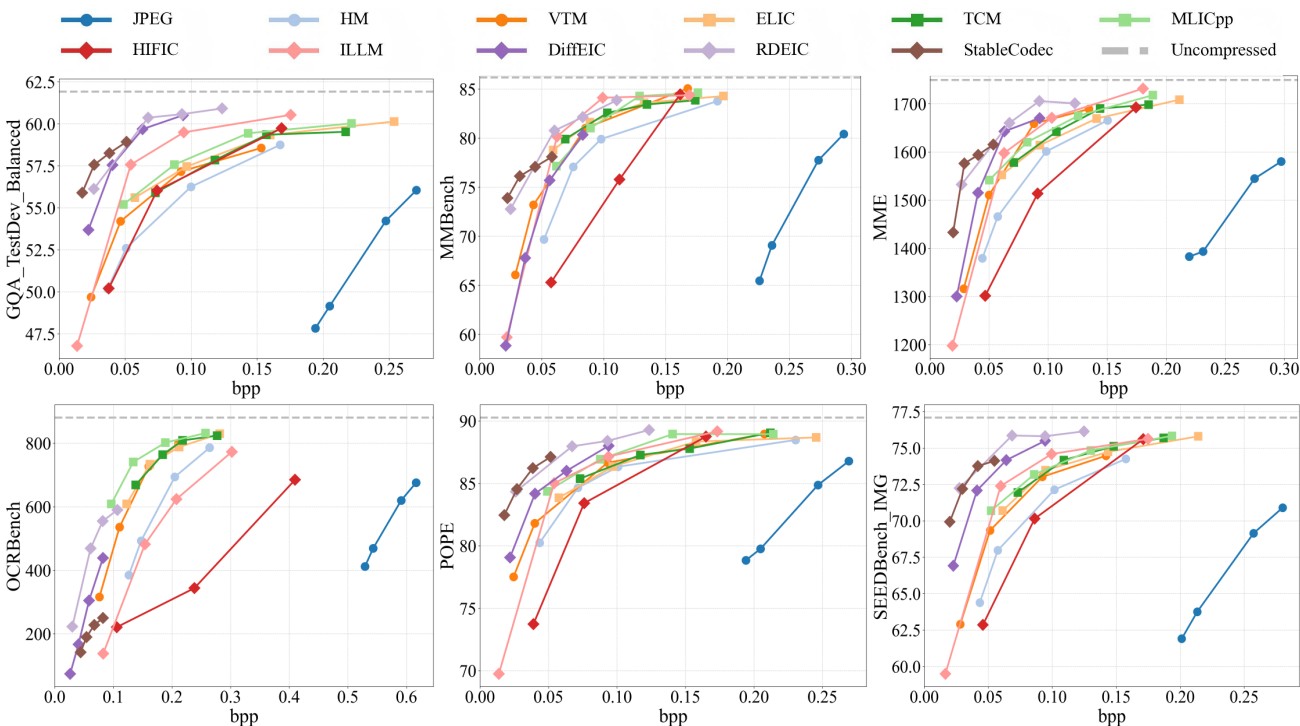

*Figure 16.* Rate-Metric curves for all types of codecs on GQA, MMB, MME, OCRBench, POPE, and SEEDBench using InternVL3-8B.

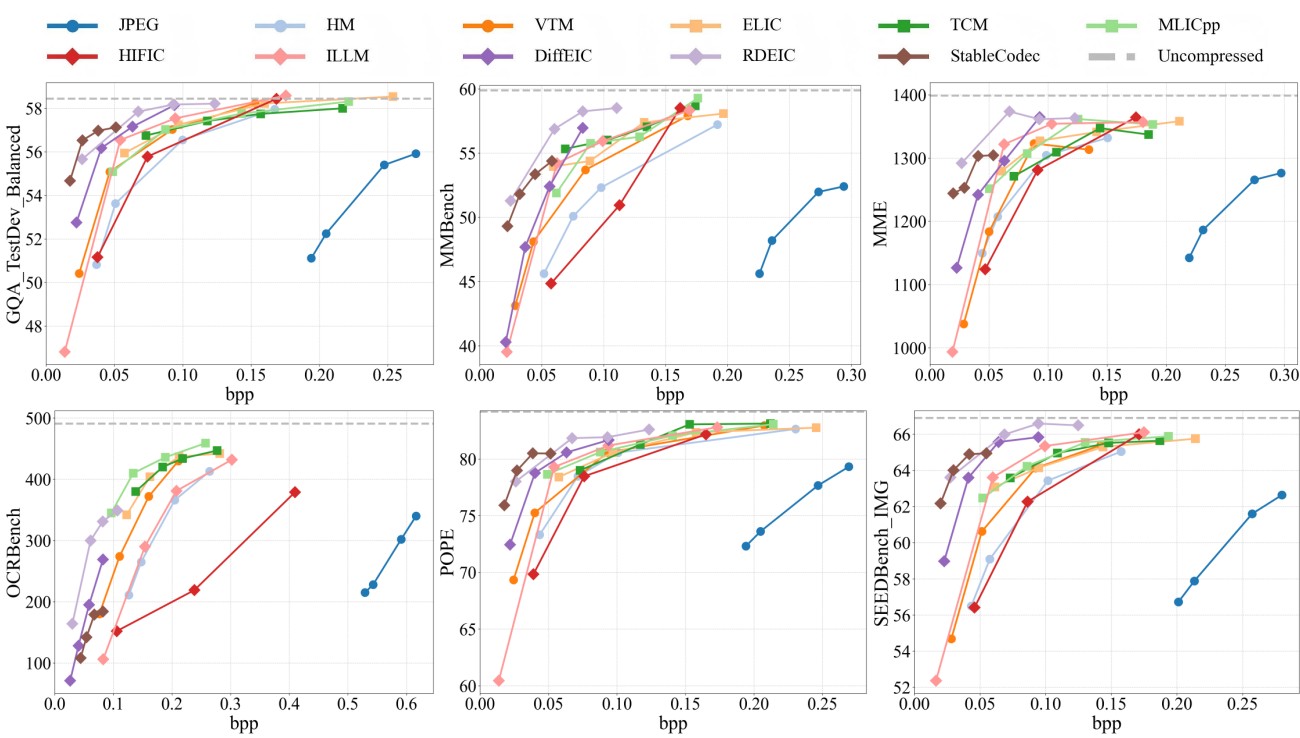

*Figure 17.* Rate-Metric curves for all types of codecs on GQA, MMB, MME, OCRBench, POPE, and SEEDBench using Janus-Pro-1B.

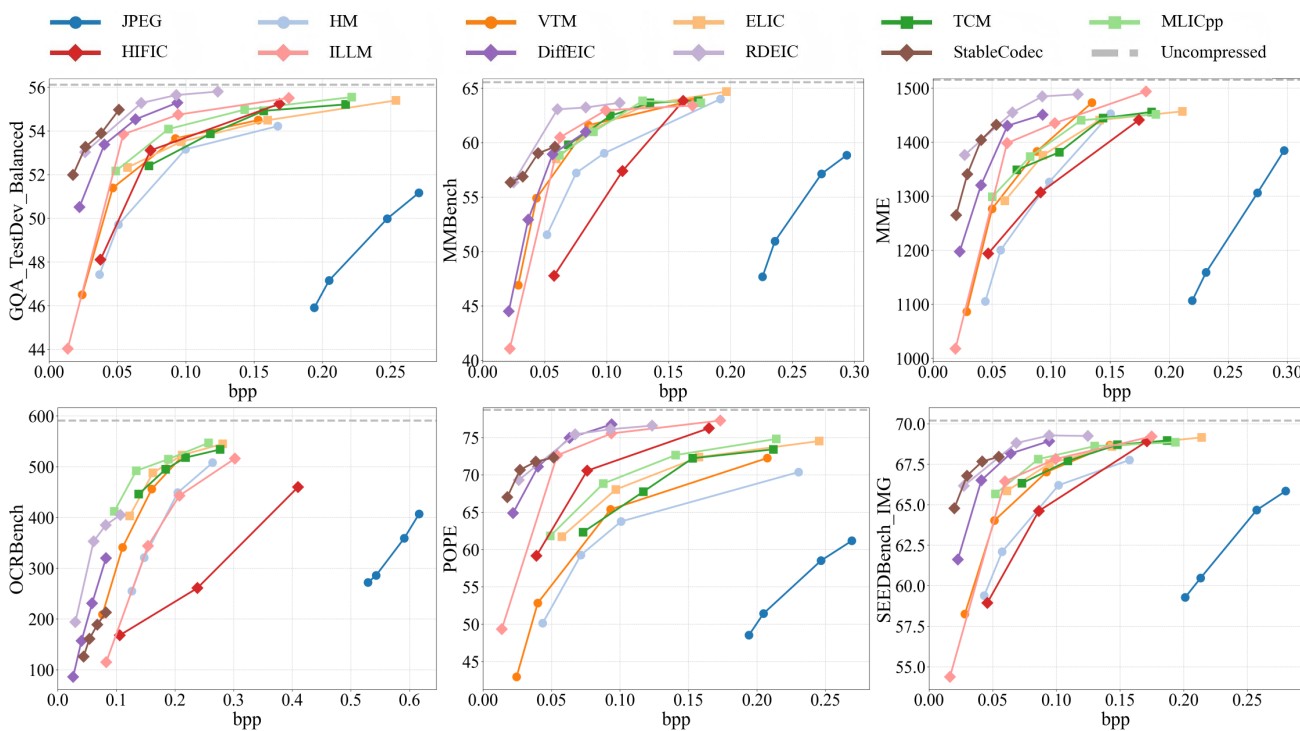

*Figure 18.* Rate-Metric curves for all types of codecs on GQA, MMB, MME, OCRBench, POPE, and SEEDBench using Janus-Pro-7B.

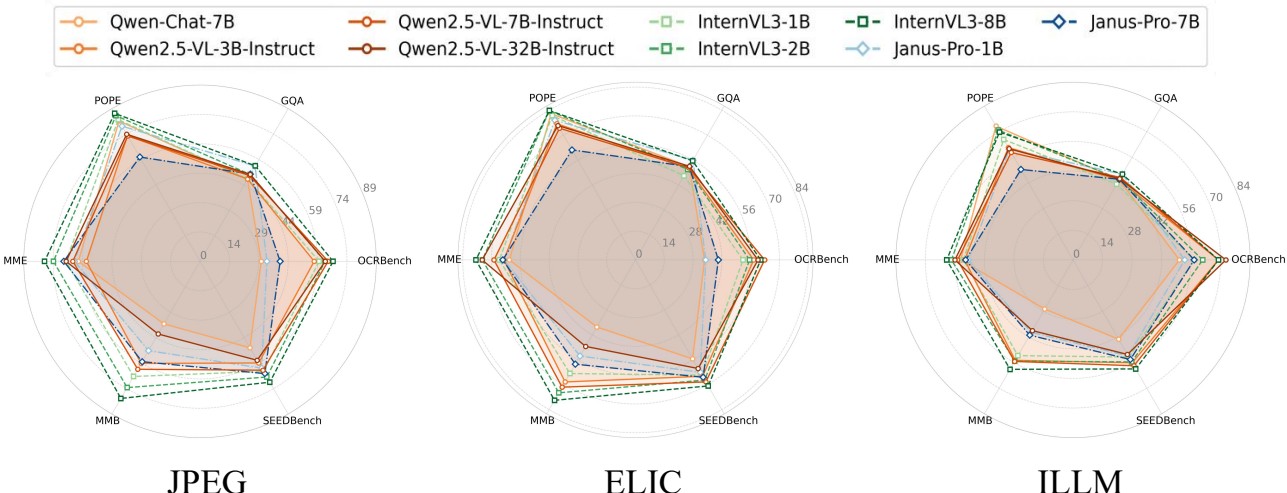

JPEG            ELIC            ILLM

*Figure 19.* Visualization of various VLM models across all metrics under three different compression distortion conditions.

*Table 17.* Comparison of BD-Metrics for different VLMs across various tasks. The value measures the average decrease in metrics relative to the uncompressed condition under the same compression bitrate for different codecs. Red fonts indicate the best-performing models, while blue fonts denote the second-best.

| VLM | Metric | JPEG | HM | VTM | ELIC | TCM | MLIC | HiFiC | MSILLM | D.EIC | RDEIC | S.Codec |
|---|---|---|---|---|---|---|---|---|---|---|---|---|
| Qwen VL2.5 -3B | OCRB | -339.1 | -242.4 | -251.1 | -98.9 | -86.2 | -87.0 | -485.0 | -308.0 | -576.0 | -387.6 | -624.8 |
| | GQA | -14.25 | -7.63 | -7.01 | -4.05 | -3.78 | -4.07 | -5.89 | -3.99 | -3.64 | -1.97 | -3.88 |
| | POPE | -17.75 | -6.16 | -6.59 | -4.61 | -4.68 | -4.81 | -10.95 | -7.80 | -7.34 | -5.44 | -8.13 |
| | MME | -506.9 | -342.2 | -309.6 | -153.7 | -142.1 | -139.2 | -126.2 | -182.4 | -190.0 | -78.0 | -183.6 |
| | MMB | -27.95 | -9.01 | -8.03 | -4.82 | -4.65 | -4.66 | -13.23 | -6.62 | -12.70 | -5.61 | -8.20 |
| | SEEDB | -19.08 | -8.38 | -7.83 | -3.90 | -3.69 | -3.85 | -7.55 | -5.10 | -3.96 | -1.85 | -3.86 |
| Intern VL3 -2B | OCRB | -314.6 | -228.0 | -242.4 | -116.1 | -104.2 | -106.5 | -483.5 | -325.4 | -597.8 | -395.4 | -639.3 |
| | GQA | -9.50 | -5.97 | -5.72 | -3.72 | -3.73 | -3.51 | -4.86 | -3.52 | -2.87 | -1.89 | -3.70 |
| | POPE | -8.28 | -3.40 | -3.61 | -2.45 | -2.38 | -2.46 | -5.98 | -4.61 | -4.67 | -2.50 | -4.76 |
| | MME | -230.8 | -151.6 | -143.9 | -60.6 | -65.8 | -60.9 | -137.8 | -105.3 | -117.4 | -51.7 | -121.3 |
| | MMB | -12.07 | -5.74 | -6.50 | -3.43 | -3.09 | -3.94 | -11.86 | -5.62 | -12.79 | -4.92 | -8.60 |
| | SEEDB | -11.62 | -6.87 | -6.30 | -3.28 | -3.35 | -3.60 | -7.21 | -4.95 | -4.12 | -2.19 | -4.48 |
| Intern VL3 -1B | OCRB | -303.7 | -226.7 | -236.6 | -110.6 | -101.5 | -104.3 | -458.3 | -321.7 | -571.9 | -381.1 | -609.2 |
| | GQA | -10.51 | -7.25 | -7.08 | -4.64 | -4.29 | -4.44 | -4.28 | -3.40 | -2.96 | -1.91 | -3.66 |
| | POPE | -11.38 | -5.15 | -6.14 | -3.96 | -3.98 | -3.59 | -7.69 | -5.88 | -5.95 | -3.25 | -6.09 |
| | MME | -231.9 | -142.5 | -153.1 | -75.4 | -68.4 | -69.0 | -137.5 | -96.1 | -119.9 | -56.7 | -143.6 |
| | MMB | -10.94 | -6.20 | -5.64 | -4.33 | -3.65 | -4.30 | -9.88 | -5.08 | -10.94 | -4.75 | -7.16 |
| | SEEDB | -10.30 | -6.09 | -5.92 | -3.23 | -3.09 | -2.91 | -5.90 | -4.27 | -3.64 | -1.83 | -4.02 |
| Janus -pro -1B | OCRB | -217.5 | -151.3 | -160.3 | -71.8 | -65.9 | -66.2 | -251.6 | -163.2 | -314.7 | -196.2 | -330.3 |
| | GQA | -4.24 | -2.33 | -2.04 | -0.57 | -0.88 | -0.99 | -2.17 | -1.92 | -1.82 | -0.81 | -1.82 |
| | POPE | -8.00 | -3.45 | -3.90 | -2.53 | -2.22 | -2.61 | -5.67 | -5.05 | -4.69 | -2.82 | -4.66 |
| | MME | -164.6 | -116.7 | -138.3 | -62.7 | -75.6 | -63.8 | -128.4 | -90.6 | -123.1 | -42.7 | -119.2 |
| | MMB | -9.38 | -6.49 | -6.15 | -3.47 | -3.17 | -3.69 | -9.68 | -5.60 | -9.15 | -3.29 | -7.30 |
| | SEEDB | -6.77 | -4.37 | -4.20 | -2.00 | -1.80 | -1.94 | -4.33 | -3.21 | -2.53 | -1.08 | -2.60 |
| GPT-4o | OCRB | -276.8 | -208.8 | -199.7 | -83.2 | -74.4 | -91.6 | -422.0 | -347.6 | -506.1 | -329.0 | -556.6 |

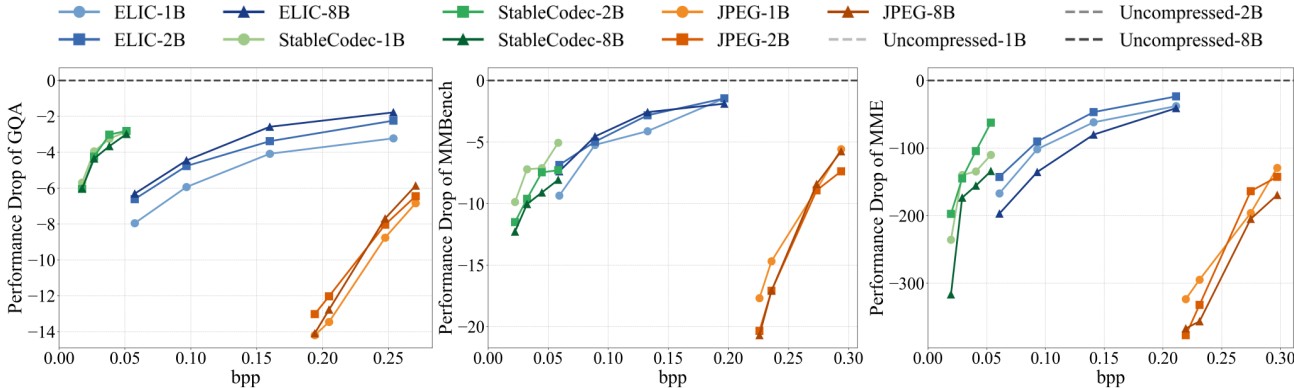

*Figure 20.* Rate-Metric drop curves to validate the scaling law based on InternVL3 series models.

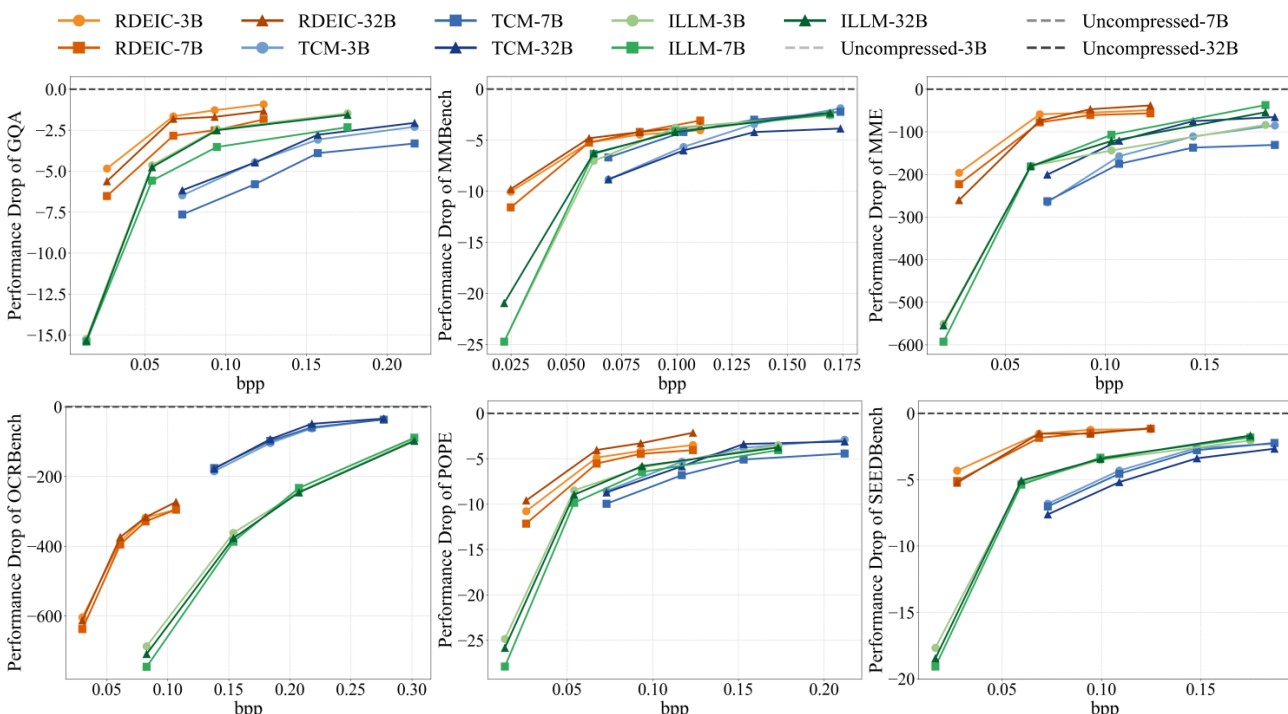

*Figure 21.* Rate-Metric drop curves to validate the scaling law based on Qwen-VL2.5 series models.

# D. More Enhancement Results

We have supplemented our experiments with a comprehensive set of additional benchmarks, including GQA, MMBench, OCRBench and MME. As shown in Figure 22, the additional results consistently confirm the effectiveness of our adapter across all evaluated settings, demonstrating its robustness to diverse compression distortions and validating its claim as a generalizable solution.

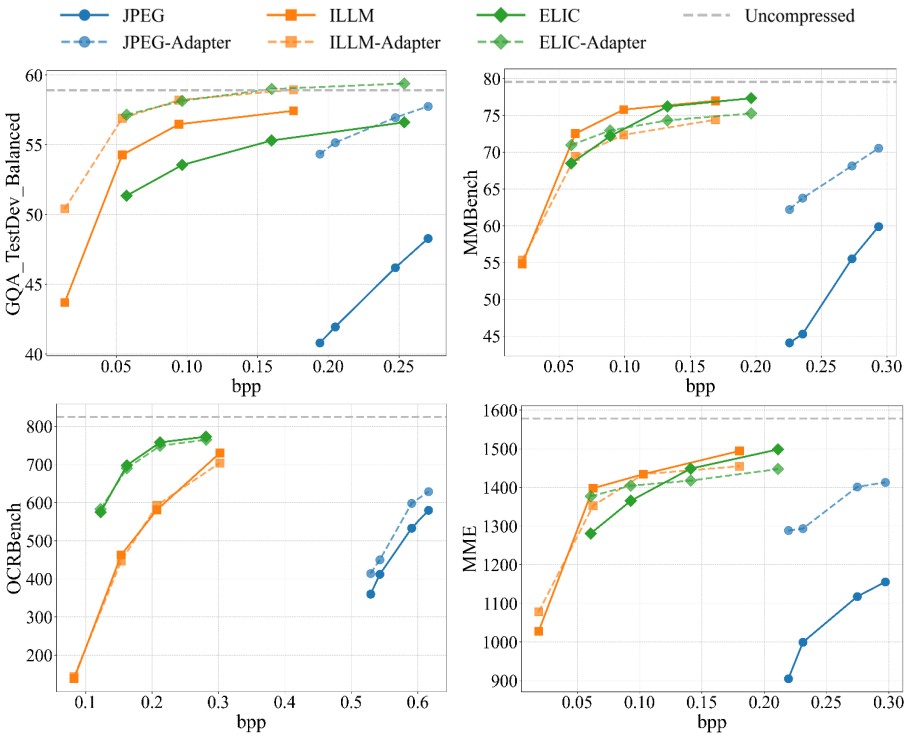

*Figure 22.* Rate-accuracy comparison on GQA, MMBench, OCRBench and MME using QwenVL2.5-3B model.

# E. Connection with Image Coding for Machines

The optimization objective of ICM tasks, under a fixed task network, follows the $R + \lambda D$ formulation. This problem can be defined as achieving higher-quality feature reconstruction at lower bitrates, which is equivalent to transmitting more effective information under a rate constraint. This corresponds to what we define as the information gap. Existing ICM methods are typically designed for specific codecs and rely on end-to-end training, where different values of $\lambda$ require training separate models. However, they do not enhance the generalization ability of the task network (VLM) itself. Therefore, we regard ICM as a task that primarily addresses the information gap.

For the TransTIC method, we followed its overall network structure and replaced the Detectron2 FPN network, originally designed for detection and segmentation tasks, with the vision encoder (VE) of Qwen-VL2.5. The training was constrained by the formulation:

$$R + \lambda D(\text{VE}(x), \text{VE}(\hat{x})), \tag{6}$$

where $D$ denotes the MSE loss. We used the same COCO dataset and experimental settings as in this paper to ensure accurate evaluation. We do not consider TransTIC to be a direct baseline for comparison, but rather an algorithm that can be further integrated and optimized in a cumulative manner. This further demonstrates the effectiveness of our proposed concepts of the generalization gap and the information gap. Even though our method was not specifically trained on TransTIC images, its generalization ability allows us to treat them as ELIC inputs and still achieve significant gains. The table below presents the parameter counts, showing that existing ICM methods indeed fine-tune far fewer parameters, but they require training separate models for different bitrates. In contrast, our approach trains only a single model, which consistently yields improvements across multiple codecs and bitrates, including ELIC, JPEG, ILLM, MLICpp, HM, DiffEIC, and TransTIC. This strongly demonstrates the importance of generalization.

Additionally, we Since their released model is only based on CLIP, which differs substantially from our work using Qwen2.5-VL, we re-trained their method for a fair comparison. The results show that our method outperforms their d1 (pre-trained for human) and d2 (updated for joint human and machine), but not their **d3** version (updated for machine), since d3 jointly optimizes both the codec and the VLM vision encoder, which can be viewed as addressing both the information gap and the generalization gap discussed in our paper. This also supports that our proposed decomposition is valid and meaningful. Nevertheless, their approach requires training a separate model for each bitrate point, whereas our method is a unified enhancement framework that generalizes across different codecs and bitrate levels. Therefore, the two works address the problem from different perspectives, and we believe both are meaningful and valuable.

