# OpenReview forum: "Benchmarking and Enhancing VLM for Compressed Image Understanding"
_ICML.cc/2026/Conference — ICML 2026 regular_

### Official Review · Reviewer_zGnq · 2026-02-24

**Soundness:** 2
**Presentation:** 2
**Significance:** 2
**Originality:** 2
**Overall Recommendation:** 4
**Confidence:** 5

**Summary:**

This paper introduces a comprehensive benchmark to evaluate Vision-Language Models (VLMs) under image compression, spanning several codecs (traditional, learned, generative), seven popular multimodal tasks, several VLMs, and over one million compressed images. The paper also proposes an adapter for the vision encoder that injects a codec-and-bitrate embedding into positional encodings and distills features from clean images.

**Compliance With Llm Reviewing Policy:**

Affirmed.

**Final Justification:**

My concerns have been properly addressed.

**Key Questions For Authors:**

- How is the codec/bpp embedding injected into RoPE in practice (additive vs multiplicative, per-layer vs first-layer only, shared vs per-block)?

 - What is the adapter’s training overhead?

**Limitations:**

yes

**Strengths And Weaknesses:**

# Strengths

1. The main strength is the breadth of the experimental study, covering a wide range of MLLMs and compression codecs.

2. This large-scale evaluation offers useful community insights into how image compression affects MLLM performance.

# Weaknesses

1. The proposed adapter is not demonstrated on the diffusion-based codecs, which are increasingly popular and may have different characteristics.

2. For the adapter performance, it lacks the comparison with SOTA method (Bridging compressed image latents and multimodal large language models， ICLR 2025)

3. The “information gap” is defined using an idealized max over model parameters. In practice, estimates rely on limited fine-tuning and may not reflect a true irreducible bound. This makes the decomposition sensitive to optimization budget and model capacity.

4. The benchmark is not evaluated on the latest frontier VLMs (e.g., GPT-5 and Gemini 3), which limits the strength of conclusions about current state-of-the-art robustness.

5. Because the benchmark focuses on a fixed set of tasks, the reported findings may not generalize well to unseen tasks or broader real-world scenarios.

---

> ### Author Rebuttal · Authors · 2026-03-31
>
> Thank you for your insightful comments. We address each of your points in detail as follows:
>
> ### W1 Demonstration on the diffusion codec
>
> In the original manuscript, we selected one representative codec from each of the three major categories, with ILLM as the generative codec. We fully agree that investigating diffusion codecs is necessary. Therefore, we expand Table 4 to include the diffusion-based codec **StableCodec**, and the experimental results show that our method consistently improves performance across all benchmarks.
>
> |Metric|POPE|SEEDB|GQA|MMB|OCRB|MME|
> |-|-|-|-|-|-|-|
> |StableCodec|2.87|0.63|1.34|0.09|1.30|3.18|
>
> ### W2 The comparison with SOTA method
>
> Our method targets the generalization gap from the VLM side rather than codec optimization, so we did not originally compare with SOTA VCM-oriented codecs. We have cited the ICLR 2025 work multiple times and it proposes ELIC-based optimization for MLLMs. We agree that a comparison is important. Since their released model is only based on CLIP, which differs substantially from our work using Qwen2.5-VL, we re-trained their method for a fair comparison. The results show that our method outperforms their **d1** (pre-trained for human) and **d2** (updated for joint human and machine), but not their **d3** version (updated for machine), since d3 jointly optimizes both the codec and the VLM vision encoder, which can be viewed as addressing both the information gap and the generalization gap discussed in our paper. This also supports that our proposed decomposition is valid and meaningful. Nevertheless, their approach requires training a separate model for each bitrate point, whereas our method is a unified enhancement framework that generalizes across different codecs and bitrate levels. Therefore, the two works address the problem from different perspectives, and we believe both are meaningful and valuable.
>
>
> |Codec|ELIC|d1|d2|d3|Ours-ELIC|
> |-|-|-|-|-|-|
> |POPE|0.00|1.34|2.67|3.92|3.42|
> |MMBench|0.00|1.01|1.83|2.62|2.45|
>
> ### W3 Both gap estimates are inaccurate.
>
> In fact, we believe that the rate-distortion bounds for specific semantic tasks are generally **intractable** to compute accurately. Precisely for this reason, we use this decomposition as an empirical framework to distinguish between two practically meaningful sources of performance degradation. Although the exact gaps are intractable, the decomposition allows us to empirically estimate a **lower bound** of the Generalization Gap, which in turn provides an **upper bound** of the residual Information Gap under a given model and optimization setup. We also make substantial efforts to ensure that training is as sufficient as possible, including careful tuning of optimization settings, so as to approximate the two gaps as closely as possible in practice, making the analysis useful.
>
> ### W4 Evaluation on the latest frontier VLMs
>
> In our work, we primarily evaluate the latest codecs and open-source VLMs, since the cost of querying commercial API-based models is prohibitively high. However, in Appendix C, we also include a comparison with the latest closed-source **ChatGPT-4o**, and observe conclusions **consistent** with those obtained from open-source models. It is reasonable to expect that similar conclusions would also hold for the latest closed-source models. Moreover, closed-source models cannot be used in our adaptor-based enhancement experiments, which makes it infeasible to verify the effectiveness of the proposed method.
>
> ### W5 The reported findings may not generalize well
>
> We have evaluated both coarse-grained and fine-grained metrics across six tasks involving more than one million compressed images, surpassing all prior studies on compression methods for VLMs in terms of both the number of tasks and the diversity of codecs. Consequently, we consider our evaluation to be comprehensive and substantial.
>
> ### Q1 Detailed information for embedding injected into RoPE
>
> From the implementation, the codec condition is incorporated through an additive conditioning mechanism. Concretely, the codec label is first transformed into a one-hot representation and mapped into the latent space by a shared embedding module, and the resulting codec embedding is added to the rotary positional embedding. Hence, the conditioning acts as an additive bias on the positional encoding. Moreover, the conditioning is introduced only once before the stack of transformer blocks, during the construction of positional embeddings. The conditioned positional embedding is subsequently reused by all blocks, indicating that the codec conditioning is shared across layers. More detailed information and ablation results can be found in our response to **mXyE**.
>
> ### Q2 Adapter's training overhead
>
> For training, we use four NVIDIA A100 GPUs in parallel and images are randomly cropped into $336\times336$ patches. We train the model with a batch size of 24 for 100k iterations, using an initial learning rate of $1\times10^{-4}$.

---

> > ### Author Rebuttal · Reviewer_zGnq · 2026-04-03
> >
> > I thank the authors for their detailed responses. While some concerns have been partially addressed, I still have remaining concerns
> >
> > - The addition of StableCodec results is appreciated. However, the reported improvements are much smaller (e.g., MMB only +0.09) compared with the numbers in Table 4 for traditional and learning-based codecs. This suggests the adapter's effectiveness may be codec-dependent and less effective on diffusion-based codecs, which are increasingly important in practice.
> > - Testing only ChatGPT-4o on a single task (OCRBench) in the appendix is insufficient to claim that conclusions transfer to frontier models. Since the authors were able to evaluate ChatGPT-4o, the marginal cost of additionally testing one or two more recent frontier models seems manageable and would substantially strengthen the benchmark's claims.
> > - Being more comprehensive than prior work is commendable but does not fully address the concern. The benchmark covers a fixed set of established VLM tasks that primarily involve short textual outputs. Tasks more directly relevant to compression, such as fine-grained visual grounding or broader document understanding scenarios, are absent. The paper's framing suggests a comprehensive benchmark, but the scope is narrower than implied.

---

> > > ### Author Response · Authors · 2026-04-07
> > >
> > > We sincerely thank the reviewer for their valuable feedback. To address all the concerns raised, we have provided further clarification and conducted additional experiments.
> > >
> > > ### W1 Explanation for the diffusion codec
> > >
> > > We also observe that although diffusion-based models achieve improvements across all tasks, their gains are relatively limited compared to other methods, especially traditional codecs and learning-based approaches.  We attribute this to the fact that existing methods are more robust to blurred textures than to the semantic distortions introduced by generative codecs. While diffusion-based codecs have recently become a research focus in image compression, no work has yet explored generative codecs specifically designed for VLM vision.  We plan to further investigate enhancement methods specifically tailored to diffusion models in future work.
> > >
> > > ### W2 Testing recent frontier models
> > >
> > > We fully agree that evaluating recent frontier VLMs can substantially strengthen the benchmark’s claims. Therefore, we have added experimental results for GPT-5-nano and GPT-5. We continue to conduct evaluations on the OCRBench dataset due to its relatively small size, although the experiments still cost several hundred dollars. The uncompressed OCRBench scores are 742 for GPT-5-nano and 808 for GPT-5. We evaluated all codecs mentioned in the paper and selected several representative ones from different categories, reporting their corresponding bpp and performance metrics in detail, as shown in the table below. The GPT-5 results outperform most of the VLMs reported in the original paper, but are weaker than QwenVL2.5-7B, which achieves a score of 884 on uncompressed images. This observation is also consistent with our Finding: **stronger VLMs generally perform better on compressed images**.
> > >
> > > |JPEG| | |TCM| | |HiFiC| | |RDEIC| | |StableCodec| | |
> > > |-|-|-|-|-|-|-|-|-|-|-|-|-|-|-|
> > > |bpp|GPT5-Nano|GPT5|bpp|GPT5-Nano|GPT5|bpp|GPT5-Nano|GPT5|bpp|GPT5-Nano|GPT5|bpp|GPT-5-Nano|GPT-5|
> > > |0.62|505|634|0.28|708|786|0.41|526|662|0.11|465|599|0.08|221|315|
> > > |0.59|450|590|0.22|675|760|0.28|273|389|0.08|445|557|0.07|205|286|
> > > |0.54|363|460|0.18|644|744|0.24|254|369|0.06|413|515|0.05|174|252|
> > > |0.52|299|436|0.14|559|682|0.11|171|258|0.03|208|284|0.04|146|191|
> > >
> > > In addition, we measure the BD-Metric of all codecs relative to the uncompressed results, as shown in the table below. The experimental results indicate that, in fine-grained OCR tasks, learning-based methods still achieve the best performance, which is consistent with the results reported in Table 2 of the original paper. We also observe that GPT-5 exhibits relatively smaller performance degradation compared to GPT-4o and other open-source VLMs, suggesting that GPT-5 is **more robust** to compression than the other models.
> > >
> > > |VLM|JPEG|HM|VTM|ELIC|TCM|MLIC|HiFiC|ILLM|DiffEIC|RDEIC|StableCodec|
> > > |-|-|-|-|-|-|-|-|-|-|-|-|
> > > |QwenVL2.5-7B|-334.6|-231.8|-241.9|-86.5|**-82.3**|-83.8|-509.9|-317.3|-601.4|-399.6|-662.1|
> > > |InternVL3-8B|-319.9|-239.7|-243.6|-117.1|**-102.1**|-107.5|-495.6|-333.5|-615.4|-408.5|-670.1|
> > > |GPT-4o|-301.9|-270.9|-238.9|-104.2|**-80.7**|-106.4|-419.1|-373.8|-552.2|-315.9|-569.0|
> > > |GPT5-Nano|-324.7|-214.8|-221.9|-91.1|-85.5|**-84.0**|-453.8|-363.9|-497.7|-343.9|-550.2|
> > > |GPT5|-269.7|-180.7|-192.3|-70.8|**-58.3**|-65.5|-405.8|-333.6|-489.8|-304.8|-537.6 |
> > >
> > > ### W3 Additional task for broader document understanding
> > >
> > > We fully agree with the reviewer’s suggestion and have added a broader document understanding fine-grained task, DocVQA. We conduct evaluation on the validation set, which contains 5,349 questions and 1,286 images, covering diverse font styles and recognition tasks, and exceeding the scale of OCRBench. We evaluate representative codecs on InternVL3-1B and QwenVL2.5-3B, and the experimental results are shown in the table below.
> > >
> > > |JPEG| | |VTM| | |TCM| | |ILLM| | |RDEIC| | |
> > > |-|-|-|-|-|-|-|-|-|-|-|-|-|-|-|
> > > |bpp|InternVL3|QwenVL2.5|bpp|InternVL3|QwenVL2.5|bpp|InternVL3|QwenVL2.5|bpp|InternVL3|QwenVL2.5|bpp|InternVL3|QwenVL2.5|
> > > |0.24|78.85|91.09|0.10|79.56|92.43|0.14|80.38|92.67|0.11|80.10|92.44|0.08|77.31|90.83|
> > > |0.23|78.58|90.45|0.07|78.76|91.50|0.10|79.68|92.09|0.07|77.91|90.37|0.06|76.59|90.19|
> > > |0.21|75.54|87.71|0.05|74.50|86.58|0.09|79.53|91.44|0.04|72.94|85.21|0.04|74.14|88.07|
> > > |0.20|73.59|85.33|0.02|61.67|73.05|0.06|78.14|89.87|0.01|14.66|17.54|0.02|41.23|53.75|
> > >
> > > Building on this, we further compute the corresponding BD-Metric values, as shown in the table below. The results indicate that learning-based codecs (TCM and ELIC) achieve better performance, while generative models like ILLM and RDEIC perform worse. This is consistent with our paper, which states that **generative models generally underperform on fine-grained text-related tasks**.
> > >
> > > |VLM|JPEG|HM|VTM|ELIC|TCM|ILLM|RDEIC|
> > > |-|-|-|-|-|-|-|-|
> > > |InternVL3|-2.69|-3.44|-4.65|-0.69|**-0.50**|-10.90|-9.25|
> > > |QwenVL2.5|-3.71|-3.95|-5.30|**-1.13**|-1.35|-12.68|-8.96|

---

### Official Review · Reviewer_kyL3 · 2026-03-07

**Soundness:** 3
**Presentation:** 3
**Significance:** 3
**Originality:** 3
**Overall Recommendation:** 4
**Confidence:** 3

**Summary:**

This paper introduces a benchmark to evaluate the robustness of Vision-Language Models (VLMs) for compressed image understanding. The authors generate compressed images with multiple quality levels using a wide range of codecs, including traditional codecs, learning-based codecs, and generative codecs. The benchmark evaluates VLM performance across fine-grained, coarse-grained, and comprehensive tasks, and studies the robustness of VLMs ranging from 1B to 32B parameters.

To improve VLM robustness to compressed images, the authors propose a lightweight adaptor that injects codec-conditioned information into the VLM encoder. The adaptor is trained via a distillation strategy, encouraging the encoder outputs for compressed images to match those produced by clean images.

**Compliance With Llm Reviewing Policy:**

Affirmed.

**Final Justification:**

Thank you for your response, i raise my score to weak accept.

**Key Questions For Authors:**

1. In Figure 5, PSNR appears to have a strong correlation with most task performances. However, the paper claims that perceptual metrics are more correlated with coarse-grained tasks. Could the authors provide a more detailed analysis explaining why perceptual metrics are considered more relevant in this setting?

2. Table 5 evaluates the adaptor on only two tasks. Since the benchmark contains multiple tasks (fine-grained, coarse-grained, and comprehensive), it would be useful to understand whether the adaptor consistently improves performance across all tasks. Could the authors provide additional evaluation results or analysis on more tasks?

3. The experiments mainly focus on relatively low bitrate regimes. From an image compression perspective, such extremely low bitrates often lead to heavy distortions that significantly affect both human and machine perception. Could the authors discuss how their findings extend to higher bitrate regimes that are more common in practical applications?

**Limitations:**

1. Although the benchmark itself covers a wide range of tasks and codecs, the proposed adaptor is evaluated on a relatively small subset of tasks. This makes it difficult to fully assess the general effectiveness of the method across the entire benchmark.

2. The experiments focus on relatively low bitrate settings, which often produce severe distortions. In many real-world scenarios, images are compressed at higher bitrates where visual quality is significantly better. Therefore, the conclusions drawn from the current experiments may not fully generalize to more typical compression settings.

**Strengths And Weaknesses:**

I would like to note that my background is mainly in image compression, and I am very familiar with the design and evaluation of image codecs. However, I am less familiar with the details of VLM architectures and evaluations. Therefore, my comments primarily focus on the compression-related aspects of the paper.

### 1. Strength
1.1 The paper includes a large variety of image codecs, tasks, and VLM models, which makes the experimental study comprehensive and credible.

1.2 The proposed VLM adaptor is lightweight and effective, and it does not require retraining the entire model, which makes it practical for deployment.

### 2. Weakness
2.1 According to Figure 5, PSNR appears to show a strong correlation with almost all evaluated tasks. However, the paper claims that perceptual metrics exhibit stronger correlations for coarse-grained tasks. From the figure, this claim is not very clear to me, as PSNR seems to correlate well with most task performance metrics. It would be helpful if the authors could provide additional clarification or analysis to support this statement.

2.2 In Table 5, the proposed adaptor is evaluated on only two tasks. Considering that the benchmark itself includes multiple tasks across different levels of difficulty, evaluating the adaptor on a broader set of tasks would provide stronger evidence of its general applicability.

2.3 From the perspective of image compression, the bitrate range used in the experiments seems relatively low. Images compressed to such low bitrates often suffer from severe distortions, and even humans may struggle to understand them. In many practical scenarios, images are usually compressed at higher bitrates. Therefore, it would be helpful if the authors could also evaluate the models at higher bitrate ranges to better reflect realistic usage scenarios.

---

> ### Author Rebuttal · Authors · 2026-03-31
>
> Thank you for your careful evaluation. We have offered clarifications and answers below:
>
> ### W1 Detailed analysis explaining for Finding 5 about PSNR
>
> Image reconstruction metrics are naturally correlated with VLM task performance, since both reflect the impact of distortion on downstream understanding. In particular, PSNR is positively correlated with task performance because higher PSNR indicates better reconstruction quality, whereas LPIPS, DISTS, and FID are negatively correlated because lower values indicate better perceptual quality. Therefore, in Figure 5, PSNR appears in red while LPIPS, DISTS, and FID appear in blue; however, the actual strength of correlation should be judged by the absolute value of the correlation coefficient.
>
> More specifically, on OCRBench, PSNR achieves a correlation coefficient of 0.97, which is substantially higher than those of LPIPS, DISTS, and FID. **In contrast, for several coarse-grained tasks, including MMBench, MME, SeedBench, and GQA, DISTS or FID shows the largest absolute correlation coefficient among all the quality metrics.** Based on these results, we concluded that perceptual metrics exhibit stronger correlations for coarse-grained tasks.That said, we also note that POPE is an exception, where PSNR still shows the strongest correlation, although its coefficient is below 0.9. We therefore agree that our current wording could be made more precise.
>
> ### W2 Additional evaluation results or analysis on more tasks for Table 5
>
> Our main performance comparison is already reported in Table 3, where we present results on all six tasks in the benchmark. Due to space limitations, Figure 5 only shows part of the results, while the complete set of curves is provided in Figure 22 of Appendix D.
>
> Table 5 was intended as a supplementary generalization experiment, so we only included two representative tasks in the main paper. However, we fully agree that evaluating the adaptor on a broader set of tasks would provide a more complete picture of its effectiveness. **Therefore, we expand Table 5 to include results on all task categories**:
>
>
> |VLM| |Qwen VL2.5-3B| | |Intern VL3-1B| |
> |-|-|-|-|-|-|-|
> |Codec|HM|MLICpp|DiffEIC|JPEG|ELIC|ILLM|
> |POPE|2.98|3.32|2.15|8.36|2.19|2.45|
> |SEEDB|3.12|1.22|-0.16|5.62|1.19|0.42|
> |MME|130.6|50.0|-3.13|133.1|25.6|7.26|
> |OCRB|2.10|5.73|10.51|3.93|6.75|5.64|
> |GQA|5.48|2.01|-0.37|8.58|4.17|2.24|
> |MMB|1.25|2.52|1.20|1.40|0.86|0.73|
>
>
> Our additional results show a trend that is consistent with the original findings: the adaptor provides stable improvements across different task types, which further supports the general effectiveness of the proposed method.
>
> ### W3  Evaluating the models at higher bitrate ranges
>
> From the perspective of machine vision-oriented image compression, very high bitrates are often unnecessary, because **machine perception typically does not require pixel-level fidelity to the same extent as human perception**, and instead focuses more on preserving semantic information. In many practical scenarios, such as 24/7 camera systems, low-bitrate storage and transmission are especially important. For this reason, **prior works in this area commonly focus on low-bitrate regimes, typically below 0.4 bpp**, such as [1] and [2].
>
> In our manuscript, Figure 3 in the original paper demonstrates that, within this bitrate range, the performance is already approaching that of lossless compression. **We further provide experimental result on high bitrate regime ($\ge 0.4$ bpp).** It is shown that the performance of compressed images does not differ too much from uncompressed images. Therefore, we think that investigating compression at low bitrates is of greater significance.
>
>
> | |bpp|POPE|bpp|SEEDBench|
> |-|-|-|-|-|
> |Uncompress|-|86.21|-|73.80|
> |JPEG|0.743|83.75|0.714|72.81|
> |ELIC|0.579|85.42|0.489|72.99|
> |ILLM|0.477|84.79|0.458|72.97|
>
> [1] Video Coding for Machine: Compact Visual Representation Compression for Intelligent Collaborative Analytics, TPAMI 2024
>
> [2] Bridging compressed image latents and multimodal large language models, ICLR 2025

---

> > ### Author Rebuttal · Reviewer_kyL3 · 2026-04-03
> >
> > Thank you for your response, i will adjust my score accordingly.

---

> > > ### Author Response · Authors · 2026-04-07
> > >
> > > Thank you very much for your thoughtful and constructive feedback on our manuscript. We are glad that we have addressed all your concerns, and we greatly appreciate your positive and encouraging comments, as well as your indication that you would consider raising your score from the previous score 3. We will carefully revise the paper based on your valuable suggestions.

---

### Official Review · Reviewer_zLfT · 2026-03-12

**Soundness:** 3
**Presentation:** 3
**Significance:** 2
**Originality:** 2
**Overall Recommendation:** 4
**Confidence:** 3

**Summary:**

This paper introduces a comprehensive benchmark for VLM agianst compressed images. The authors categorize performance loss into two dimensions: the Information Gap (physical loss of semantic details) and the Generalization Gap (model inability to handle compression artifacts). To mitigate both gaps, a universal VLM adaptor is proposed.

**Compliance With Llm Reviewing Policy:**

Affirmed.

**Final Justification:**

After carefully reviewing all of the feedback, I have decided to maintain my initial recommendation of "weak accept." While I agree with Reviewer mXyE that the impact on the field may be limited, the paper successfully addresses a common, everyday scenario that has been previously overlooked.

**Key Questions For Authors:**

Please refer to weakness

**Limitations:**

yes

**Strengths And Weaknesses:**

Strength:

- The paper identifies a significant bottleneck for the deployment of VLMs in bandwidth-constrained environments, providing a timely benchmark for the community.


- The decomposition of performance degradation into "Information Gap" and "Generalization Gap" provides a structured way to analyze why multimodal models fail under distortion.


Q1. The authors define the "Information Gap" as the performance loss that remains after fine-tuning. However, this definition is tautological and dependent on the optimization strategy. The residual loss could stem from suboptimal training, insufficient data diversity, or the specific architecture of the adaptor rather than an inherent "physical limit" of the compressed data. Without an information-theoretic analysis (e.g., rate-distortion bounds for semantic tasks), the distinction between the "Generalization Gap" and "Information Gap" remains empirically fuzzy.

Q2. If I understand correclty, the "Universal VLM Adaptor" relies on manual input of the codec type m and compression level n via one-hot encoding. This assumes that the compression parameters are known at inference time. In real-world scenarios, images are often transmitted without such metadata. A truly "universal" solution should include a blind degradation estimation module; as currently designed, the adaptor's performance is bound to the accuracy of external metadata, limiting its practical "universality."

Typo: In Table 1 caption, representative is mispelled as “representa(t)ive” .

---

> ### Author Rebuttal · Authors · 2026-03-31
>
> Thank you for your insightful review. We have provided detailed responses as follows:
>
> ### W1 The visualization of "Generalization Gap" and "Information Gap" remains fuzzy.
>
> In fact, we believe that the rate-distortion bounds for specific semantic tasks are generally **intractable** to compute accurately. Precisely for this reason, we use this decomposition as an empirical framework to distinguish between two practically meaningful sources of performance degradation: the portion that can be recovered through adaptation to compressed inputs, and the portion that remains unrecovered due to compression-induced distortion. Although the exact gaps are intractable, the decomposition allows us to empirically estimate a **lower bound** of the Generalization Gap, which in turn provides an **upper bound** of the residual Information Gap under a given model and optimization setup. This makes the analysis useful even if it is not theoretically exact.
>
> In our implementation, we instantiate this framework using the VLM vision encoder and define the adaptation objective such that the semantic features extracted from compressed images are aligned with those from uncompressed images. We also make substantial efforts to ensure that training is as sufficient as possible, including careful tuning of optimization settings, so as to approximate the two gaps as closely as possible in practice. We fully acknowledge that the resulting estimates still depend on the training strategy, as the reviewer pointed out. However, we believe that such approximations are still valuable. They provide an operational characterization of the two gaps and corresponding empirical bounds, which can serve as a useful reference point for future research on compression-aware VLM adaptation.
>
> ### W2  The adaptor's performance is bound to external metadata, limiting its practical universality.
>
> We agree that the practical applicability of the adapter depends on whether codec-related side information is available at inference time, and we would like to clarify this point from two perspectives. On the one hand, our conditioning assumption is aligned with many practical multimedia transmission and storage pipelines. In standard compressed image/video bitstreams, codec-related information such as **the codec format** (e.g. H. 264, H. 265, H. 266) and **quantization parameters** (e.g., QP) is typically available as part of the bitstream syntax or accompanying metadata, since such information is required to support correct decoding. Therefore, in many deployment scenarios, assuming access to codec type and compression level is reasonable rather than overly restrictive.
>
> On the other hand, we fully agree that blind open-world settings, where such metadata is unavailable, are also highly relevant. To examine this case, we additionally performed a no-condition ablation study as shown in following Table, including **removing all conditional components** (*w./o.* all meta), **using only distortion-level conditions** (*w./o.* codec type meta), and **using only codec conditions** (*w./o.* distortion meta). The results confirm that the unconditioned variant remains effective. However, we also observed an undesirable phenomenon in the no-condition setting: as the bitrate increases, the adapted VLM performance may instead decrease. A plausible explanation is that, without explicit side information, the adapter tends to bias its learning toward more severely distorted samples, which can hurt its behavior on higher-bitrate images with milder degradation. To address this issue, we further introduced the distortion level condition into the adapter. This effectively alleviates the above problem and leads to more stable and improved performance across different compression levels. Building on this, we found that additionally incorporating the codec identity provides further gains, since different codecs introduce different artifact patterns and distortion characteristics.
>
> |Codec|Metric|*w./o.* all meta|*w./o.* codec type meta|*w./o.* distortion meta|Ours|
> |-|-|-|-|-|-|
> |JPEG|POPE|11.86|12.22|12.43|12.62|
> | |SEEDB|11.01|11.41|12.54|12.88|
> |ELIC|POPE|2.91|3.07|3.28|3.42|
> | |SEEDB|0.44|0.45|0.62|0.69|
> |ILLM|POPE|3.16|3.19|3.41|3.52|
> | |SEEDB|1.14|1.18|1.21|1.23|

---

> > ### Author Rebuttal · Reviewer_zLfT · 2026-04-03
> >
> > I appreciate the authors' clarification, particularly the no-condition ablation study, which strengthens the paper's claims. Despite the missing SOTA comparisons mentioned by other reviewers, the motivation of the work is sound. Therefore, I am keeping my rating as Weak Accept.

---

> > > ### Author Response · Authors · 2026-04-07
> > >
> > > Thank you very much for your positive opinion and valuable comments! We are glad that our clarification, especially the no-condition ablation study, has helped strengthen the paper's claims. In the revised version, we will carefully add and discuss relevant comparisons to further improve the paper.

---

### Official Review · Reviewer_mXyE · 2026-03-13

**Soundness:** 3
**Presentation:** 3
**Significance:** 3
**Originality:** 3
**Overall Recommendation:** 5
**Confidence:** 4

**Summary:**

This work presents a detailed study of downstream visual understanding performance of large VLMs under compressed input images. Motivated by the lack of detailed studies under this domain, the work performs comprehensive evaluation with Qwen, InternVL and Janus models on an extensive set of multimodal understanding benchmarks and provides insights on the capability degradation of these models under different image compression methods. The authors further group this degradation into an irreducible "information gap" and a reducible "generalization gap" and propose an adapter mechanism to bridge the gap for the latter.

**Compliance With Llm Reviewing Policy:**

Affirmed.

**Final Justification:**

Though I do not believe that this work would have enormous impact on one or more sub-fields of AI, I still believe that it demonstrates comprehensive evaluation with multiple models widely used in the literature. The proposed adapter mechanism likewise requires some polishing (either in the engineering or in the writing front), though I do not believe that it is in a state that would necessitate a sure rejection.

I believe that the insights provided by the work are broadly interesting, and I am in the opinion that more people in the community would think so.

Accordingly, I am raising my score to a clear acceptance (4 -> 5) along with my confidence in this review (3 -> 4).

**Key Questions For Authors:**

- Can the authors comment a bit more on the methodological aspects of the adapter, in particular on how they arrived at the exact design choices, whether they have ablated over other options and whether they can think of ways to make it more generalizable across codecs/compression rates?

**Limitations:**

Yes.

**Strengths And Weaknesses:**

With respect to soundness, presentation, significance and originality, the work has the following strengths and weaknesses:

- **Soundness**: The work's primary claims are two-folds, (i) a comprehensive evaluation of recent and well-known large VLMs for multimodal understanding tasks under compressed image settings and (ii) an adapter mechanism to bring their performance closer to the case where the images are not compressed.
  - The first of these claims is well-supported throughout the work. The authors benchmark 9 well-known large VLMs with an appropriate choice of models on 11 different image codecs and on 7 tasks covering coarse captioning (COCO-Captions) and OCR-heavy tasks.
  - The findings of these evaluations are also interesting and sound, with one finding highlighting the problem, two studying how much degradation happens at different scales and another two finding demonstrating which codecs seem to be better for large VLMs and how the performance correlates with how humans perceive.
  - However, there are some drawbacks of the proposed adapter solution, especially for practical deployment.  The primary drawback stems from the fact that the method injects an embedding conditioned on the oracle knowledge of the codec and its precise compression level at inference-time. For open-world applications, such metadata may not always be available and could hinder the applicability of this remedy.
  - Finally, providing more ablations on the design choices associated with the adapter, e.g. removing the conditional part in the RoPE, just to isolate the effect of each and every component would be a nice-to-have.

- **Presentation**: The overall language of the work is easy-to-follow, the figures are neatly drawn and the tables are clearly organized. Presentation also mostly reflects the actual contributions of the work without inflating or conflating them. One area of improvement here could be if the authors were to elaborate more on the design choices associated with the adapter mechanism verbally and detail the exact rationale behind the core design choices.

- **Significance**:  The broader domain of applications studied by the work is highly relevant for applications of large VLMs on several important domains, such as web agents and on-edge applications.
  - The findings of the comprehensive evaluation represent a valuable resource for the community working in these domains.
  - However, the significance of the proposed adapter mechanism is constrained by the aforementioned soundness concerns. In particular, the proposed method relies on oracle knowledge of the codec and compression level.

- **Originality**: To the best of my knowledge, such a comprehensive evaluation of many novel VLMs across a wide-array of image codecs is novel. However, algorithmically, the originality of the proposed adapter is limited and it serves more like an ad-hoc adaptation of existing methods into this particular domain,

---

> ### Author Rebuttal · Authors · 2026-03-31
>
> We sincerely appreciate your interest in our work. We have provided detailed responses as follows:
>
> ### W1 The adaptor's performance is bound to external metadata
>
> We agree that the practical applicability of the adapter depends on codec-related metadata at inference time, and we would like to clarify this point from two perspectives. On the one hand, our conditioning assumption is aligned with many practical multimedia transmission and storage pipelines. In standard compressed image/video bitstreams, codec-related information such as the **codec format** (e.g. H. 264, H. 265, H. 266) and **quantization parameters** (e.g., QP) is typically available as part of the bitstream syntax or accompanying metadata, since such information is required to support correct decoding. Therefore, in many deployment scenarios, assuming access to codec type and compression level is reasonable rather than overly restrictive. On the other hand, we fully agree that blind open-world settings, where such metadata is unavailable, are also highly relevant. To examine this case, we additionally performed no-condition ablations as shown in W3, in which the adapter does not receive codec/compression-level information. The results confirm that explicit conditioning is beneficial when metadata is available, **but the unconditioned variant remains effective, demonstrating that the proposed remedy still has value beyond the fully informed setting.**
>
> ### W2 Detailed information for the adapter
>
> To enable the vision encoder to explicitly account for compression artifacts, we incorporate codec-related information, including the distortion type and compression level as conditional input. Specifically, assuming there are $m$ codec types and each codec has $n$ compression levels, we first represent the codec identity and compression level using one-hot encoding, and then project them into the latent space through an embedding function $T(\cdot)$ composed of two linear layers. This yields the codec condition embedding $C_{\mathrm{emb}}$, defined as:
> $$
> C_{\mathrm{emb}} = T(m, n, d)
> $$
> where $d$ indicates the embedding dimension aligning with the VLM vision encoder.  For Qwen2.5-VL, we set $d=40$. To inject the codec condition into all spatial positions of the visual tokens, we draw inspiration from the fusion strategy of conditional embeddings and time embeddings in conditional diffusion models. We apply RoPE to the dimensions $h$ and $w$ obtained after the input image is divided into patches, resulting in an initial positional embedding of dimension $d$ and add the codec condition $C_{\mathrm{emb}}$ to the RoPE representation, obtaining the conditional positional embedding $P_{\mathrm{emb}}$.
> $$
> P_{\mathrm{emb}} = \mathrm{RoPE}(h, w, d) + C_{\mathrm{emb}}.
> $$
> The codec condition is injected once at the positional-encoding stage, rather than being separately injected inside each transformer block. The resulting conditioned positional embeddings are then reused by all subsequent blocks. In this way, codec-aware information is injected into the positional encoding and made available to the VLM throughout the image.
>
> ### W3 More ablations on the design choices associated with the adapter
>
> We have added three additional ablation settings to better analyze the design choices of the adapter: **removing all conditional components** (*w./o.* all meta), **using only distortion-level conditions** (*w./o.* codec type meta), and **using only codec conditions** (*w./o.* distortion meta), as summarized in the table below.
>
> |Codec|Metric|*w./o.* all meta|*w./o.* codec type meta|*w./o.* distortion meta|Ours|
> |-|-|-|-|-|-|
> |JPEG|POPE|11.86|12.22|12.43|12.62|
> | |SEEDB|11.01|11.41|12.54|12.88|
> |ELIC|POPE|2.91|3.07|3.28|3.42|
> | |SEEDB|0.44|0.45|0.62|0.69|
> |ILLM|POPE|3.16|3.19|3.41|3.52|
> | |SEEDB|1.14|1.18|1.21|1.23|
>
> Interestingly, we found that even without any condition information, the adapter can still provide clear performance gains for VLMs. This indicates that the proposed method remains effective even in the absence of explicit metadata, which is encouraging for more practical blind settings. However, we also observed an undesirable phenomenon in the no-condition setting: **as the bitrate increases, the adapted VLM performance may instead decrease**. A plausible explanation is that, without explicit side information, the adapter tends to bias its learning toward more severely distorted samples, which can hurt its behavior on higher-bitrate images with milder degradation. To address this issue, we further introduced the distortion level condition into the adapter. This effectively alleviates the above problem and leads to more stable and improved performance across different compression levels. Building on this, we found that additionally incorporating the codec identity provides further gains, since different codecs introduce different artifact patterns and distortion characteristics. This motivates the final adapter design adopted in our paper.

---

> > ### Author Rebuttal · Reviewer_mXyE · 2026-04-03
> >
> > I acknowledge the rebuttal of the authors and thank them for their efforts in trying to address my concerns.
> >
> > I can clearly state that most of my concerns are resolved and I find the work to be interesting. Furthermore, I do not believe that the work lacks merit to be accepted at this venue and the other lacking points (e.g. points raised by other reviewers) are more extensions than items necessary to make this work acceptable.
> >
> > Accordingly, I will be raising my score to reflect this.
> >
> > Finally, I believe that the adapter is still a weakness of the work in its current format and I highly encourage the authors to either integrate the discussion items they have raised in their rebuttal with more detail to the final manuscript or significantly tone down their claims regarding the precise contributions on that front.

---

> > > ### Author Response · Authors · 2026-04-07
> > >
> > > Thank you for your positive and constructive feedback! We are glad that all of your concerns have been resolved. Regarding the adapter, we will carefully follow your suggestion in the revision and add more detailed discussion from the rebuttal. Thank you again for your valuable guidance.

---

### Decision · Program_Chairs · 2026-04-30

**Decision:**

Accept (regular)

**Comment:**

This paper presents a comprehensive benchmark for evaluating VLM models under compressed image settings, and proposes an adapter-based method to improve robustness across different codecs and compression levels. The work further analyzes performance degradation through a decomposition into information gap and generalization gap, providing empirical insights into model behavior under compression.

The paper received overall positive evaluations after rebuttal, and reviewers generally agreed that the benchmark is good and the problem is timely and relevant. The rebuttal addressed most concerns, and scores were updated accordingly.

That said, several points raised during the review process would benefit from further clarification and strengthening in the final version. First, regarding the proposed adapter, reviewers noted that its reliance on codec and compression-level metadata may limit its applicability in open-world settings. While the rebuttal provided additional analysis, it would be helpful to more clearly discuss this assumption, its practical implications, and possible extensions toward blind or metadata-free scenarios. In addition, further clarification of the adapter design choices and more detailed ablations would improve transparency and reproducibility.

Then, the decomposition into information gap and generalization gap, while intuitively appealing, remains largely empirical. Reviewers pointed out that this formulation may depend on optimization and training choices, and would benefit from a more careful discussion of its limitations and interpretation, or stronger theoretical grounding where possible.

Overall, this is a solid and well-executed piece of work with clear value to the community. The above points are intended as suggestions to further improve clarity, rigor, and impact in the final version.